# Nucleophile sensitivity of *Drosophila* TRPA1 underlies light-induced feeding deterrence

Eun Jo Du[1,2†], Tae Jung Ahn[1,2†], Xianlan Wen[1,3†], Dae-Won Seo[1,4,5†], Duk L Na[1,4,5†], Jae Young Kwon[6], Myunghwan Choi[7,8], Hyung-Wook Kim[9], Hana Cho[1,3], KyeongJin Kang[1,2*]

[1]Samsung Biomedical Research Institute, Seoul, Republic of Korea; [2]Department of Anatomy and Cell Biology, Sungkyunkwan University School of Medicine, Suwon, Republic of Korea; [3]Department of Physiology, Sungkyunkwan University School of Medicine, Suwon, Republic of Korea; [4]Department of Neurology, Samsung Medical Center, Sungkyunkwan University School of Medicine, Seoul, Republic of Korea; [5]Neuroscience Center, Samsung Medical Center, Seoul, Republic of Korea; [6]Department of Biological Sciences, Sungkyunkwan University, Suwon, Republic of Korea; [7]Department of Biomedical Engineering, Sungkyunkwan University, Suwon, Republic of Korea; [8]Center for Neuroscience Imaging Research, Institute for Basic Science, Suwon, Republic of Korea; [9]College of Life Sciences, Sejong University, Seoul, Republic of Korea

*For correspondence: kangk@ skku.edu

[†]These authors contributed equally to this work

Competing interests: The authors declare that no competing interests exist.

**Abstract** Solar irradiation including ultraviolet (UV) light causes tissue damage by generating reactive free radicals that can be electrophilic or nucleophilic due to unpaired electrons. Little is known about how free radicals induced by natural sunlight are rapidly detected and avoided by animals. We discover that *Drosophila* Transient Receptor Potential Ankyrin 1 (TRPA1), previously known only as an electrophile receptor, sensitively detects photochemically active sunlight through nucleophile sensitivity. Rapid light-dependent feeding deterrence in *Drosophila* was mediated only by the TRPA1(A) isoform, despite the TRPA1(A) and TRPA1(B) isoforms having similar electrophile sensitivities. Such isoform dependence re-emerges in the detection of structurally varied nucleophilic compounds and nucleophilicity-accompanying hydrogen peroxide ($H_2O_2$). Furthermore, these isoform-dependent mechanisms require a common set of TRPA1(A)-specific residues dispensable for electrophile detection. Collectively, TRPA1(A) rapidly responds to natural sunlight intensities through its nucleophile sensitivity as a receptor of photochemically generated radicals, leading to an acute light-induced behavioral shift in *Drosophila*.

## Introduction

Highly reactive chemical compounds are dangerous because of their ability to covalently modify and incapacitate proteins and nucleic acids. Such reactive chemicals are often perceived as noxious and are avoided by animals through chemical nociception mechanisms, in which Transient Receptor Potential (TRP) channels play major roles (*Julius, 2013*). TRPA1 is the only conserved reactive chemical receptor in the bilateria from humans to flies (*Kang et al., 2010*), and is activated rather than inactivated by covalent modification unlike most proteins (*Macpherson et al., 2007*). Although chemical reactivity can be categorized into two opposite characters, electron-attracting electrophilicity and electron-donating nucleophilicity, only the former has been shown to provoke TRPA1-

**eLife digest** Atoms are made up of a nucleus that contains protons and neutrons, which is orbited by electrons. The electrons orbit within shells that surround the nucleus and each shell can contain a specific number of electrons. A particle with an outer shell that is missing one or more electrons will be unstable and highly reactive. It will attempt to achieve a full outer shell either by sharing electrons with another particle, or by donating or stealing an electron. Particles that steal electrons are said to be "electrophilic" (electron-loving) while those that donate them are "nucleophilic".

Electrophilic and nucleophilic particles can damage DNA and proteins. In species from fruit flies to humans, electrophilic substances such as formaldehyde activate a type of ion channel called TRPA1. These ion channels contribute to pain signaling, and their activation triggers unpleasant and painful sensations that deter animals from getting too close to electrophilic substances. However, it is not known if animals have an equivalent mechanism to help them avoid toxic nucleophilic compounds, like carbon monoxide and cyanide.

Du, Ahn, Wen, Seo, Na et al. now show that fruit fly neurons produce two versions of the TRPA1 channel: one that is sensitive to electrophiles, plus a second that is sensitive to nucleophiles in addition to electrophiles. The existence of nucleophile-sensitive TRPA1 helps explain why fruit flies avoid feeding in strong sunlight. Ultraviolet radiation in sunlight triggers the production of reactive forms of oxygen that behave as strong nucleophiles. These reactive oxygen species – which can damage DNA – activate the nucleophile-sensitive TRPA1 and thereby trigger the fly's avoidance behavior.

Human TRPA1 responds only to electrophiles and not to nucleophiles. By targeting the nucleophile-sensitive version of insect TRPA1, it may thus be possible to develop insect repellants that humans do not find aversive. Furthermore, TRPA1s from some insect species are more sensitive to nucleophiles than others, with a mosquitoes' being more sensitive than the fruit flies'. This means that insect repellants that target nucleophile-sensitive TRPA1 could potentially repel malaria-transmitting mosquitoes without affecting other insect species.

dependent nociception. Furthermore, there is no molecular mechanism attributed to the sensory detection of nucleophiles, while nucleophilic compounds are widespread in nature as antioxidant phytochemicals (*Lü et al., 2010*) and as decomposition gases of animal carcasses (*Dent et al., 2004*), and strong nucleophiles, such as carbon monoxide and cyanide, can be fatal to animals (*Grut, 1954*; *Krahl and Clowes, 1940*).

In insects, TRPA1 was originally thought to be a polymodal sensory receptor capable of detecting both temperature increases (*Viswanath et al., 2003*; *Hamada et al., 2008*; *Corfas and Vosshall, 2015*) and chemical stimuli (*Kang et al., 2010*; *Kwon et al., 2010*). However, this polymodality would limit reliable detection of chemical stimuli when ambient temperature varies. In fact, the *TrpA1* genes in *D. melanogaster* and malaria-transmitting *Anopheles gambiae* were recently found to produce two transcript variants with distinct 5' exons containing individual start codons (*Kang et al., 2012*). The two resulting TRPA1 channel isoforms, TRPA1(A) and TRPA1(B), differ only in their N-termini, and share more than 90% of their primary structure. TRPA1(A), which is expressed in chemical-sensing neurons, is unable to confer thermal sensitivity to the sensory neurons, allowing TRPA1(A)-positive cells to reliably detect reactive chemicals regardless of fluctuations in ambient temperature. In addition to the insufficient thermosensitivity, TRPA1(A) has been under active investigations for its novel functions, such as the detection of citronellal (*Du et al., 2015*), gut microbiome-controlling hypochlorous acid (*Du et al., 2016*), and bacterial lipopolysaccharides (*Soldano et al., 2016*). Although TRPA1(A) and TRPA1(B) are similarly sensitive to electrophiles (*Kang et al., 2012*), the highly temperature-sensitive TRPA1(B) is expressed in internal AC neurons that direct *TrpA1*-dependent long-term thermotaxis of the animal (*Hamada et al., 2008*; *Ni et al., 2013*), and is thereby inaccessible to reactive chemicals present in the environment. Thus, the functional segregation of TRPA1 isoforms into two distinct sensory circuits is critical for sensory discrimination between thermal and chemical inputs.

Photochemical conversion of photonic to chemical energy greatly affects organisms, as is evident in vision, circadian rhythm, and photosynthesis. Low-wavelength solar radiation that reaches the surface of the Earth, generally in the range of ultraviolet (UV) to blue light, is a major driving force for such natural photochemical reactions. In contrast to the beneficial effects of photochemistry, the chemical reactivity of free radicals generated by low-wavelength light imposes DNA and tissue damage (*Murphy, 1975*; *Hannan et al., 1984*) and accelerates aging (*Fisher et al., 1997*; *Gordon and Brieva, 2012*). TRPA1 has been characterized in the bilateria (*Kang et al., 2010*) as the molecular receptor for oxidative electrophilic reactivity, as reactive electrophilic compounds activate the non-selective cation channel through covalent modification of key cysteines in the ankyrin repeat domain (*Hinman et al., 2006*; *Macpherson et al., 2007*). Despite its electrophile sensitivity, mammalian TRPA1 requires an extremely high UV intensity (580 mW/cm$^2$) for direct activation (*Hill and Schaefer, 2009*), which is at least 4-fold greater than the extraterrestrial solar constant (SC: the total solar irradiation density measured by a satellite, 137 mW/cm$^2$ [*Gueymard, 2004*]). The high UV intensity requirement for TRPA1 activation in mammals indicates that electrophilic sensitivity is inadequate for sensitive detection of photochemically-produced free radicals, although radicals are often regarded as inflicting electrophilic oxidative stress. However, *Drosophila* TRPA1 has been shown to readily respond to UV and $H_2O_2$ with the physiological significance and molecular basis of its enhanced sensitivity unknown (*Guntur, 2015*).

Insects and birds are able to visualize upper-UV wavelengths (above 320 nm) via UV-specific rhodopsins (*Salcedo et al., 2003*; *Ödeen and Håstad, 2013*). Visual detection of UV in this range by insects generally elicits attraction towards the UV source rather than avoidance (*Craig and Bernard, 1990*; *Washington, 2010*). At the same time, lower UV wavelengths, such as UVB (280–315 nm) at natural intensities, have been known to decrease insect phytophagy (*Zavala et al., 2001*; *Rousseaux et al., 1998*) via a direct effect on the animals that does not involve the visual system (*Mazza et al., 1999*). However, the molecular mechanism of UV-induced feeding deterrence has yet to be unraveled. Here, using feeding assays combined with the *Drosophila* molecular genetics and electrophysiological analyses in in vivo neurons and heterologous *Xenopus* oocytes, we show that TRPA1(A) is a nucleophile receptor, and that the ability to detect nucleophilicity enables TRPA1(A) to detect light-evoked free radicals and mediate light-dependent feeding deterrence.

## Results

### UV irradiation evokes *TrpA1*-dependent action potentials in *Drosophila* i-bristle sensilla and suppresses feeding

Insect herbivory is often reduced by solar UV radiation (*Mazza et al., 1999*, *2002*; *Kuhlmann, 2009*), suggesting that UV radiation is responsible for acute control of insect feeding through a light-sensitive molecular mechanism. To examine whether UV radiation deters feeding through a direct impact on insect gustatory systems, we turned to the *Drosophila* model system. First, we tested if the aversive taste pathway responds to UV illumination using extracellular single sensillum recording, which monitors action potentials from *Drosophila* labellum taste neurons (*HODGSON et al., 1955*). Aversion to bitter chemicals is in part coded in i-bristles (*Weiss et al., 2011*), which house single bitter-tasting neurons (*Tanimura et al., 2009*). Illumination of 295 nm UV light at an intensity of 5.2 mW/cm$^2$(~85% of the total UV intensity on the ground [6.1 mW/cm$^2$]) received by the fly labellum (*Figure 1—figure supplement 1a, b, d*) rapidly elicited firing of single taste neurons in i-a bristles which was sustained after illumination (*Figure 1a, b*). Bitter-sensing taste cells in i-bristles also act as receptors for tissue-damaging chemicals through expression of the conserved reactive electrophile sensor TRPA1 (*Kang et al., 2010*; *Kang et al., 2012*). Because free radicals elicited by UV illumination are often regarded as oxidative electrophiles, we examined the i-bristles of the *TrpA1^{ins}* mutant flies, which lack a functional *TrpA1* gene (*Rosenzweig et al., 2008*; *Kang et al., 2010*; *Kang et al., 2012*). Interestingly, *TrpA1^{ins}* showed an severely reduced UV response in i-bristles, suggesting the importance of *TrpA1* for UV sensing in these sensilla (*Figure 1a,b*). The cell viability of bristles without UV responses was confirmed with 1 mM berberine (*Figure 1—figure supplement 2*), a bitter chemical that selectively excites bitter-sensing neurons in i-a bristle sensilla (*Weiss et al., 2011*). To assess whether the UV-dependent excitation of *TrpA1*–positive cells pertains to a reduction in the insects' appetite, a modified capillary feeder (Café) assay (*Ja et al., 2007*) was used to appraise the

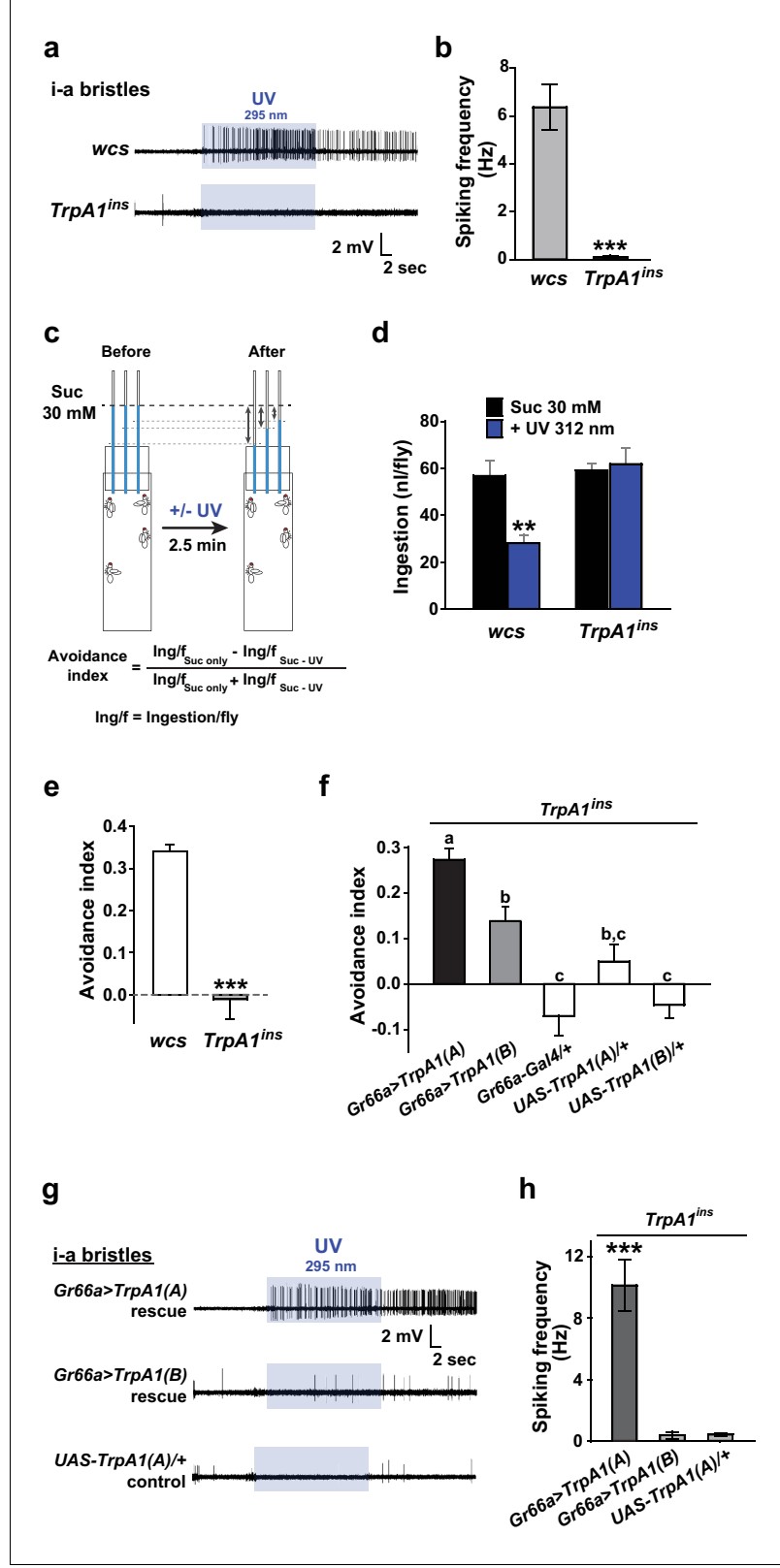

**Figure 1.** UV-induced taste neuron firing and feeding deterrence require *TrpA1(A)* but not *TrpA1(B)*. (**a**) Representative UV responses from i-bristles of the indicated genotypes. *wcs*: white canton S (wild type). *TrpA1ins*: *TrpA1* knockout flies. Recording taken under 5.2 mW/cm$^2$ UV illumination is marked by purple boxes. (**b**) Averaged data from **a** (n = 4–5). (**c**) Schematic illustration of modified Café assays used to test UV-induced feeding

*Figure 1 continued on next page*

*Figure 1 continued*

deterrence. (**d**) Ingestion amount/fly with or without 1.3 mW/cm$^2$ 312 nm UV illumination in *wcs* and *TrpA1$^{ins}$*. (**e**) Suppression of feeding by UV illumination in the indicated genotypes, presented as an 'avoidance index' (n = 5–6). (**f**) Reintroduction of transcript variants *TrpA1(A)* and *TrpA1(B)* cDNAs differentially restores UV avoidance of *TrpA1$^{ins}$*. Letters indicate statistically distinct groups (p<0.05, n = 8–12, Tukey's test). (**g**) Isoform-dependent rescue of UV-evoked neuronal responses in *TrpA1$^{ins}$*. (**h**) Summary of **g** (n = 5–6). *p<0.05, **p<0.01, ***p<0.001, Tukey's, Student's t- or Mann-Whitney U tests.

The following figure supplements are available for figure 1:

**Figure supplement 1.** Setups for electrophysiological recordings of UV-evoked responses in in vivo taste neurons and *Xenopus* oocytes and estimation of light irradiance at the illuminated tissue.

**Figure supplement 2.** Cell viability check for non-responders in extracellular recording experiments.

**Figure supplement 3.** Temperature changes in Café assays that measure UV-dependent feeding avoidance.

effect of UV on feeding. Pairwise Café experiments were conducted in two groups of overnight-starved flies, which were allowed to drink 30 mM sucrose for 2.5 min from calibrated glass capillaries inserted into a fly vial with or without 312 nm UVB illumination reaching the inside of the vial at ~1.3 mW/cm$^2$ (*Figure 1c*, see Materials and methods for details), an irradiance less than 25% of the total UV intensity received on the surface of the Earth (*RReDC*; *Gueymard, 2004*). Avoidance indices were calculated to concisely compare the gustatory effect of UV across genotypes (*Figure 1c*). UV illumination substantially decreased the feeding of wild-type flies (WT: *wcs*), but did not disrupt the feeding of *TrpA1$^{ins}$* mutants (*Figure 1d,e*), suggesting that the *TrpA1*- and UV-dependent spikings in aversive taste neurons suppress food ingestion.

## UV-evoked neuronal and behavioral responses depend on the TRPA1(A) but not the TRPA1(B) isoform

UV irradiation imposes toxicity on biological tissues by generating free radicals, which are often regarded as oxidative electrophiles. TRPA1(A) is similar to TRPA1(B) in its responsiveness to reactive electrophiles, although the two isoforms have distinct thermal sensitivities (*Kang et al., 2012*). For this reason, we anticipated that the two TRPA1 isoforms would be similar in UV responsiveness. However, reintroduction of either *TrpA1(A)* and *TrpA1(B)* cDNA to *Gr66a-Gal4* bitter cells resulted in differential restoration of UV-induced feeding deterrence in *TrpA1$^{ins}$* (*Figure 1f*). *Gr66a-Gal4* (*Dunipace et al., 2001*) drives expression in the bitter taste neurons of s- and i-bristle sensilla and has been successfully used in RNAi-knockdown and restoration of *TrpA1* in feeding behavior experiments (*Kang et al., 2010,2012*), indicating that it covers most of the *TrpA1*-positive taste neurons. In sensillum recordings, *TrpA1(B)* expression in *TrpA1$^{ins}$* i-bristle bitter neurons produced very few UV-evoked spikes, while *TrpA1(A)* expression generated robust action potentials in response to UV illumination (*Figure 1g,h* and *Figure 1—figure supplement 2*). This implies that the feeding avoidance observed with *TrpA1(B)* expression in *Figure 1f* may be due to TRPA1(B)-related temperature sensitivity but not UV sensitivity. Indeed, the 2.5-minute-long UV illumination used in the feeding assays raised the temperature in the vials by 1.6 ± 0.07°C and 3.25 ± 0.13°C with and without active cooling, respectively, at ambient temperatures between 22.5 and 23°C (*Figure 1—figure supplement 3a*). The greater extent of temperature increase in the vials without air-cooling resulted in higher avoidance in animals expressing *TrpA1(B)* but not for *TrpA1(A)*-expressing or WT animals (*Figure 1—figure supplement 3b*). These data suggest that the mild UV-dependent feeding deterrence induced by *TrpA1(B)* in *TrpA1$^{ins}$* animals resulted from the high thermosensitivity of TRPA1(B), which sensed a temperature difference of 1.6°C.

## Exogenous expression of TRPA1(A) but not TRPA1(B) in sugar-sensing neurons confers neuronal and behavioral UV responsiveness

The contrasting UV sensitivity of the two TRPA1 isoforms may be related to the intracellular environment of bitter neurons, which contain only TRPA1(A) (*Kang et al., 2012*), rather than to their

functional divergence. To test this possibility, the isoforms were ectopically expressed in the *Gr5a-Gal4* (*Marella et al., 2006*) sweet-tasting neurons. Consistent with the previously observed lack of responsiveness to TRPA1 agonists (*Kang et al., 2012*), the L-bristles of control animals failed to respond to 295 nm UV light, with the exception of a few mechanosensory responses occasionally caused by bristle deflection upon contact with the electrolyte (*Figure 2a,b*, and *Figure 1—figure supplement 2*). Similar to the results from *Gr66a-Gal4* cells, *Gr5a* neurons expressing *TrpA1(A)* but not those expressing *TrpA1(B)* showed robust UV-induced firing (*Figure 2a,b*), although, unlike bitter-sensing neurons, UV-evoked firing in *Gr5a* neurons was attenuated soon after the removal of illumination (*Figures 1g* and *2a*). Furthermore, *Gr5a-Gal4* rescue with *TrpA1(A)* and *TrpA1(B)* enabled *TrpA1^{ins}* flies to extend their proboscis in response to UV and infrared (IR) light (*Figure 2c*), demonstrating that expression of the two isoforms transforms sweet-tasting neurons into UV and IR receptors, respectively. These results suggest that TRPA1(A) is capable of responding to UV light without the co-expression of other signaling factors such as GPCRs, as the intrinsic molecular thermosensor TRPA1(B) independently reacts to IR, which warms TRPA1(B)-expressing cells (*Kang et al., 2012*).

## Heterologous expression of TRPA1(A), not TRPA1(B), in *Xenopus* oocytes provides UV-evoked current responses

To further confirm that TRPA1(A) serves as a molecular sensor of UV, we turned to *Xenopus* oocytes as a heterologous expression system. At an unnaturally high intensity of 350 nm UVA illumination, such as 580 mW/cm$^2$ (~420% of SC and >9,000% of total UV intensity on the ground), mammalian TRPA1 was directly stimulated by UV illumination when heterologously expressed (*Hill and Schaefer, 2009*). In line with the observation that *Drosophila* TRPA1(A) confers a low threshold of UV responsiveness in nonnative cells, heterologous *Xenopus* oocytes expressing TRPA1(A) but not those expressing TRPA1(B) showed TRPA1-dependent current increases in response to 295 nm UV light at ~62% of the total ground UV intensity, 3.8 mW/cm$^2$ (*Figure 2d–f*; for an estimation of UV irradiance received by oocytes see *Figure 1—figure supplement 1e*). The UV-induced current exhibited the reversal potential and outward rectification previously associated with currents recorded from fly TRPA1 (*Kang et al., 2010, 2012*). TRPA1 unselectively conducts cations, with reversal potentials close to 0 mV. The UV and NMM responses serially recorded from each cell showed similar reversal potentials of $-5.6 \pm 1.1$ and $-5.4 \pm 1.0$ mV, respectively (not significantly different, paired t-test, n = 7). Rectification was quantitated by calculating the ratio between the conductances at +60 to that at $-60$ mV. Moderate outward rectification was indicated by the ratios of the net UV and NMM responses (*Figure 2d,g*), which were $1.8 \pm 0.2$ and $1.7 \pm 0.3$ (not significantly different, paired t-test, n = 4), respectively, when TRPA1(A) showed a similar degree of activation in response to UV and NMM. Such UV-responsive ability of TRPA1(A) in frog cells indicates the sufficiency of TRPA1(A) as an autonomous UV receptor expressed in fly bitter-sensing neurons. In contrast, human TRPA1 (humTRPA1) expressed in oocytes failed to yield current responses to the UV intensity of 3.8 mW/cm$^2$ (*Figure 2—figure supplement 1*) as expected from the excessive intensity required previously (*Hill and Schaefer, 2009*). Furthermore, inside-out macropatches from TRPA1-expressing oocytes also responded to UV light in an isoform-dependent manner (*Figure 2—figure supplement 2a,b,e*). To exclude the possibility of leak current induced by UV illumination, we recorded from TRPA1(B)-containing membranes over extended periods of time (up to 350 s) and did not observe a significant increase in current. Activation of TRPA1(A) often showed a delayed onset before UV-evoked current responses, unlike TRPA1(A) in the whole-cell configuration, suggesting that cytosolic reducing power aids in UV-dependent TRPA1(A) activation. The ability to confer UV responsiveness to ectopic fly neurons and *Xenopus* oocytes strongly argues that TRPA1(A) serves as the molecular UV receptor without other upstream signaling molecules or coreceptors.

## Nucleophilicity-bearing H$_2$O$_2$ induces robust behavioral, neuronal and heterologous responses through TRPA1(A) but not TRPA1(B)

Next, we asked why TRPA1(A), but not TRPA1(B), can respond to UV light. The two isoforms differ in their N-termini which comprises less than 10% of the primary protein structure, but their reactive electrophile sensitivity is comparable (*Kang et al., 2012*). We conducted conventional Café assays to confirm the similarity of sensitivity of the isoforms to the electrophile N-methyl maleimide (NMM) (*Figure 3a*). WT but not *TrpA1^{ins}* animals showed NMM-dependent feeding avoidance as previously

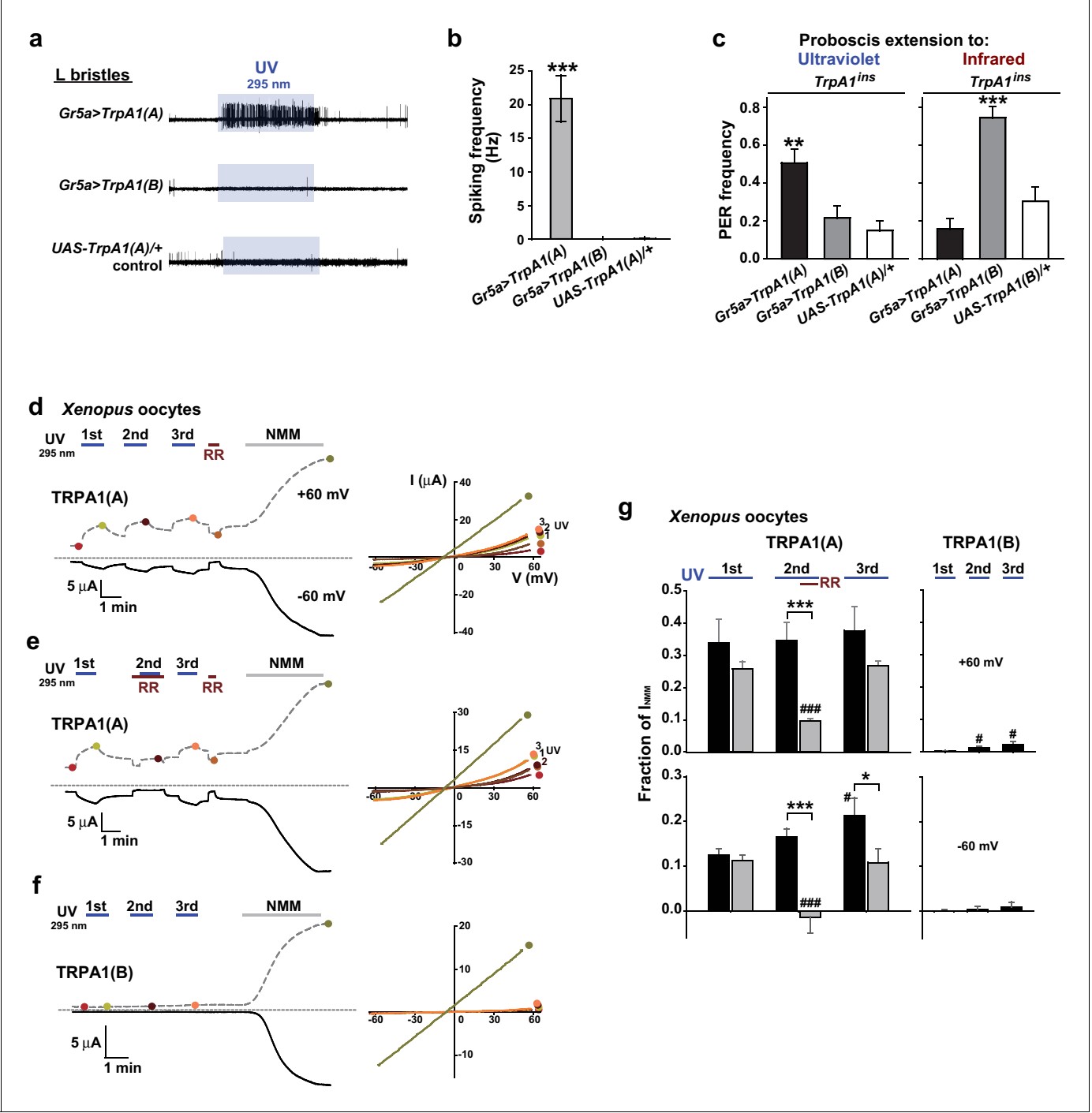

**Figure 2.** TRPA1(A) expression is sufficient for UV responsiveness. (**a**) UV- and isoform-dependent action potentials in sweet-sensing neurons exogenously expressing TRPA1 isoforms. (**b**) Summary of **a** (n = 4–5). (**c**) Proboscis extension reflex (PER) to UV (n = 24–25) and IR (n = 22–24) in *TrpA1*[ins] flies ectopically rescued in sweet taste neurons. (**d-f**) Typical UV-evoked currents in *Xenopus* oocytes expressing the indicated isoforms. RR: 0.2 mM ruthenium red. NMM: 0.1 mM. *Right*, Current-voltage (IV) relationships at the indicated points in the *Left* panels. (**g**) Summary of **d–f**. UV responses normalized to NMM currents at +60 and −60 mV, respectively (n = 4–5). #: $p < 0.05$, ###: $p < 0.001$, ANOVA Repeated Measures test compared to the first response (n). *$p < 0.05$, **$p < 0.01$, ***$p < 0.001$, Tukey's, Student's t- or Mann-Whitney U tests.

The following figure supplements are available for figure 2:

**Figure supplement 1.** Human TRPA1 (humTRPA1) is not activated by the same UV intensity as *Drosophila* TRPA1(A).

Figure 2 continued

**Figure supplement 2.** TRPA1(A)s from flies and mosquitoes do not need the cytosol of *Xenopus* oocytes for UV responsiveness.

reported (*Kang et al., 2012, 2010*). The reintroduction of either *TrpA1(A)* or *TrpA1(B)* cDNA similarly restored NMM-dependent feeding avoidance in *TrpA1$^{ins}$*, demonstrating that the isoforms are similar in their ability to confer electrophile responsiveness in vivo. This raises the possibility that TRPA1(A) detects a property of UV-generated free radicals other than oxidizing electrophilicity. Unpaired electrons in free radicals serve as both electrophiles and nucleophiles (*Domingo and Pérez, 2013*), as the lone electrons favor pairing by either accepting (electrophilic) or donating (nucleophilic) an electron. The primary oxyradical superoxide ($O_2^{\cdot-}$) (molecular oxygen that gained an electron), arising from UV illumination, is a well-known nucleophilic reductant (*Danen and Warner, 1977*). Also, hydrogen peroxide ($H_2O_2$), which can be derived from $O_2^{\cdot-}$, is not only an oxidizing electrophile but also a reducing nucleophile owing to its two key chemical properties. First, when nucleophilic atoms, such as sulfur, nitrogen and oxygen, are adjacent to each other, the nucleophilicity of the compounds is dramatically increased (the alpha effect [*Edwards and Pearson, 1962*]). $H_2O_2$ (H-O-O-H) contains two consecutive oxygen atoms, which supposedly renders it nucleophilic. Second, $H_2O_2$, a weak acid, yields the hydroperoxide anion ($HOO^-$), a strong nucleophile (*Pearson and Edgington, 1962*). To examine if TRPA1 isoforms differentially respond to $H_2O_2$, $H_2O_2$-dependent feeding avoidance was tested with Café assays. WT flies increasingly avoided ingestion of $H_2O_2$-containing food as the dose of $H_2O_2$ was increased from 10 to 100 mM, while *TrpA1$^{ins}$* did not (*Figure 3b*). The robust spiking response of bitter-sensing neurons in i-bristles to 100 mM $H_2O_2$ required the *TrpA1* gene (*Figure 3c,d*, and *Figure 3—figure supplement 1*). Like UV responses, feeding avoidance (*Figure 3e*) and neuronal responses (*Figure 3f,g* and *Figure 3—figure supplement 1*) to $H_2O_2$ were preferentially rescued by *TrpA1(A)* rather than *TrpA1(B)*. Ectopic expression in *Gr5a-Gal4* neurons recapitulated the isoform dependence observed in bitter-sensing cells (*Figure 3h,i* and *Figure 3—figure supplement 1*), indicating that the differential outcomes from expression of *TrpA1* transcript variants are unrelated to cellular context.

To date, $H_2O_2$-responding TRPs have been characterized as being indirectly stimulated and/or requiring high doses (>1 mM) of $H_2O_2$ to generate current under physiological conditions (*Yoshida et al., 2006*; *Fonfria et al., 2004*). In particular, extracellular $Ca^{2+}$ is a requisite for the moderate $H_2O_2$ sensitivity (EC50 =~ 230 µM) of $Ca^{2+}$-conducting mouse TRPA1 (*Andersson et al., 2008*), which is activated directly by an elevation in intracellular $[Ca^{2+}]$ (*Wang et al., 2008*; *Zurborg et al., 2007*), providing evidence that $H_2O_2$ is a weak electrophilic oxidant compared to other electrophilic TRPA1 agonists. Interestingly, *Drosophila* TRPA1(A) heterologously expressed in *Xenopus* oocytes was readily activated by $H_2O_2$ at concentrations as low as 100 nM (*Figure 3j,k*, EC50 = 5.0±0.8 µM, and *Supplementary file 1*). In contrast, the response of TRPA1(B) was slow and required high $H_2O_2$ concentrations (*Figure 3j,k*, EC50 = 0.9±0.2 mM), possibly because the response of TRPA1(B) depends solely on the electrophilicity of $H_2O_2$, similar to mammalian TRPA1s. The ~450-fold higher sensitivity of TRPA1(A) than TRPA1(B) in oocytes may account for the differential behavioral and neuronal $H_2O_2$ responses of the TRPA1 isoforms. Thus, $H_2O_2$ mimics UV in that feeding inhibitions by $H_2O_2$ and UV rely on *TrpA1(A)*, suggesting that the nucleophilicity of $H_2O_2$ and UV-generated radicals is critical for activation of TRPA1(A). The high $H_2O_2$ concentration required for neuronal and behavioral responses compared to that required to evoke heterologous responses in oocytes may be due to enzymatic catalysis of $H_2O_2$ in *Drosophila* taste bristles, the activity of which may not be robust in *Xenopus* oocytes. In addition, chemical doses necessary for in vivo TRPA1 activation are usually much higher than those required for activation of TRPA1 that is heterologously expressed in oocytes according to previous studies (*Kang et al., 2010, 2012*). Yet, TRPA1 isoform dependence is consistent in in vivo and in vitro studies, which demonstrates that TRPA1(A) is superior to TRPA1(B) in sensing nucleophilicity-accompanying $H_2O_2$ in various contexts.

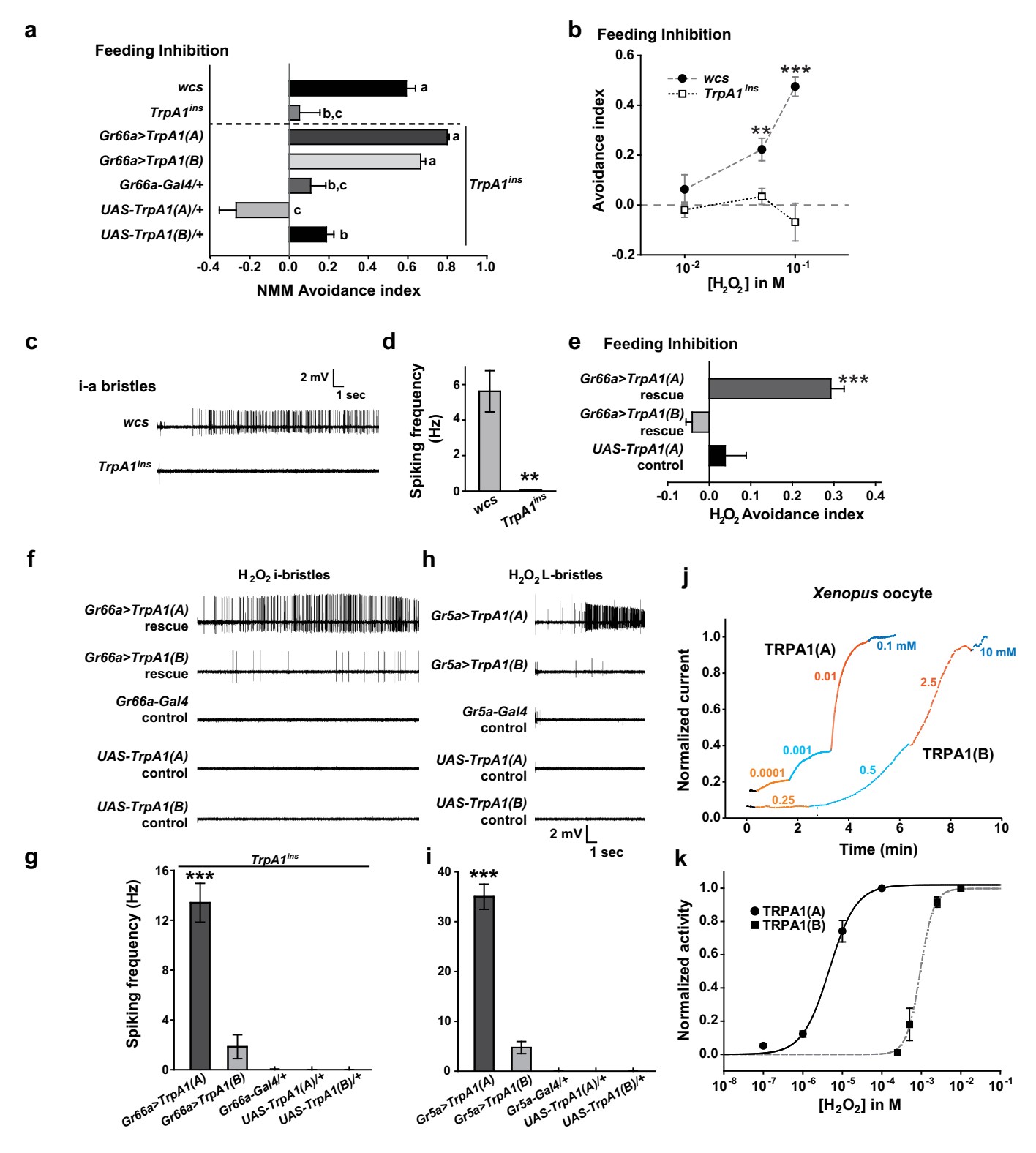

**Figure 3.** Responses of *Drosophila* TRPA1 to nucleophilicity-bearing $H_2O_2$ are isoform-dependent. (**a**) Electrophilic NMM-induced feeding avoidance is not isoform-dependent (n = 3–4). Letters indicate statistically distinct groups (p<0.01, Tukey's test). (**b–d**) Feeding deterrence (**b**) and neuronal responses (**c** and **d**) to $H_2O_2$ are abolished in *TrpA1^ins^* (b: n = 3–11, d: n = 8–14). (**e–i**) Isoform-dependent rescue of $H_2O_2$ feeding avoidance (**e**, n = 4) and neuronal activation (**f–i**, n = 6–29) in *TrpA1^ins^*. j and k, Typical $H_2O_2$ current recordings normalized to the maximum $H_2O_2$ response (**j**) and $H_2O_2$

*Figure 3 continued on next page*

*Figure 3 continued*

dose-dependence (k, n = 4–11) of TRPA1 isoforms in oocytes. Alternating colors represent increasing concentrations of H₂O₂ as indicated. **p<0.01, ***p<0.001, Tukey's or Student's t-tests.

The following figure supplement is available for figure 3:

**Figure supplement 1.** Cell viability check for non-responders in extracellular recording experiments.

## The nucleophilic reductant dithiothreitol (DTT) elicits current responses from TRPA1(A) but not TRPA1(B)

A peculiar property of TRPA1(A) is that its expression in oocytes effects small standing current at rest. This basal activity is little observed in cells expressing TRPA1(B) or the mutants TRPA1(A)C105A and TRPA1(A)R113A/R116A (*Kang et al., 2012*) (*Figure 4b,c*), in which the conserved Cys105 and Arg113/116 residues in the cytosolic N-terminus of TRPA1(A) were replaced with Ala (*Figure 4a*). This observation led to the hypothesis that the intracellular reducing/nucleophilic power for redox homeostasis partially opens TRPA1(A). To examine the idea, TRPA1 isoforms expressed in frog oocytes were subjected to perfusion buffer containing the well-known nucleophilic reductant dithiothreitol (DTT). DTT contains two nucleophilic thiols and is a popular reductant used in the studies of protein biochemistry. Indeed, only the TRPA1 channel that produced the standing current showed dose-dependent responses to DTT in oocytes (*Figure 4d*, EC50 =~92.8 μM and *Figure 4—figure supplement 1*). The DTT response of TRPA1(B) was little compared to that of TRPA1(A), revealing that detection of nucleophilic DTT by TRPA1 is also isoform-dependent. The current amplitude of TRPA1(A) evoked by H₂O₂ is intermediate between those induced by DTT and NMM; the average maximal amplitudes of DTT- and H₂O₂-evoked currents were ~10% and ~30% of NMM responses, respectively (*Figure 4—figure supplement 2*), implying that H₂O₂ synergistically stimulates TRPA1(A) through two distinct pathways.

## Mutations of conserved TRPA1(A)-specific residues that abolish DTT sensitivity compromise heterologous, neuronal, and behavioral responses to UV and H₂O₂

As mentioned above, heterologously expressed TRPA1(A)C105A and TRPA1(A) R113A/R116A in oocytes appeared to lack the constitutive activity observed with TRPA1(A)WT, suggesting that the mutants may be unable to respond to nucleophiles. Indeed, C105A and R113A/R116A substitutions compromised the DTT responsiveness of TRPA1(A) such that it was indistinguishable from that of TRPA1(B). The NMM sensitivity of these mutants was previously shown to be very similar to that of TRPA1(A)WT (*Kang et al., 2012*), indicating that the mutations specifically impaired DTT-dependent activation. Consistent with a previous study in which high concentrations of DTT completely reversed the mammalian TRPA1 current provoked by reversible electrophilic agonists (*Macpherson et al., 2007*), we found that cells expressing humTRPA1 seldom showed electrophysiological responses to DTT (*Figure 4d* and *Figure 4—figure supplement 1f*, and *Supplement file 1*). Notably, these DTT-insensitive mutants and humTRPA1 showed remarkably reduced responses to H₂O₂ (*Figure 4e,f*); the mutants and humTRPA1 were similar to TRPA1(B) in H₂O₂ sensitivity (*Supplement file 1*) and activation kinetics. These results indicate a strong structure-function association between the ability of TRPA1(A) to respond to DTT and H₂O₂ (*Figure 4e,f*). Furthermore, oocytes expressing either mutant failed to respond to 3.8 mW/cm² 295 nm UV irradiation (*Figure 4g*), revealing the concomitant requirement of the conserved residues for DTT, H₂O₂ and UV responses. To demonstrate the in vivo implications of TRPA1(A) nucleophile sensitivity in H₂O₂ and UV responsiveness, cDNAs encoding *TrpA1(A)C105A* and *TrpA1(A)R113A/R116A* were expressed in WT *Gr66a-Gal4* neurons (*Figure 4h–j*). While the *TrpA1(A)C105A* transgene was not functionally expressed in *Gr66a-Gal4* cells (*Figure 4—figure supplement 3*), expression of *TrpA1(A)R113A/R116A* in *Gr66a-Gal4* cells dramatically decreased both UV- and H₂O₂-dependent neuronal spiking responses and feeding avoidance but did not impair NMM responsiveness (*Figure 4h–j*). These data demonstrate a specific dominant negative effect of the *TrpA1(A)R113A/R116A* mutant on TRPA1(A)-mediated H₂O₂ and UV detection in vivo, and strongly support the notion that the *TrpA1*-positive neurons are necessary

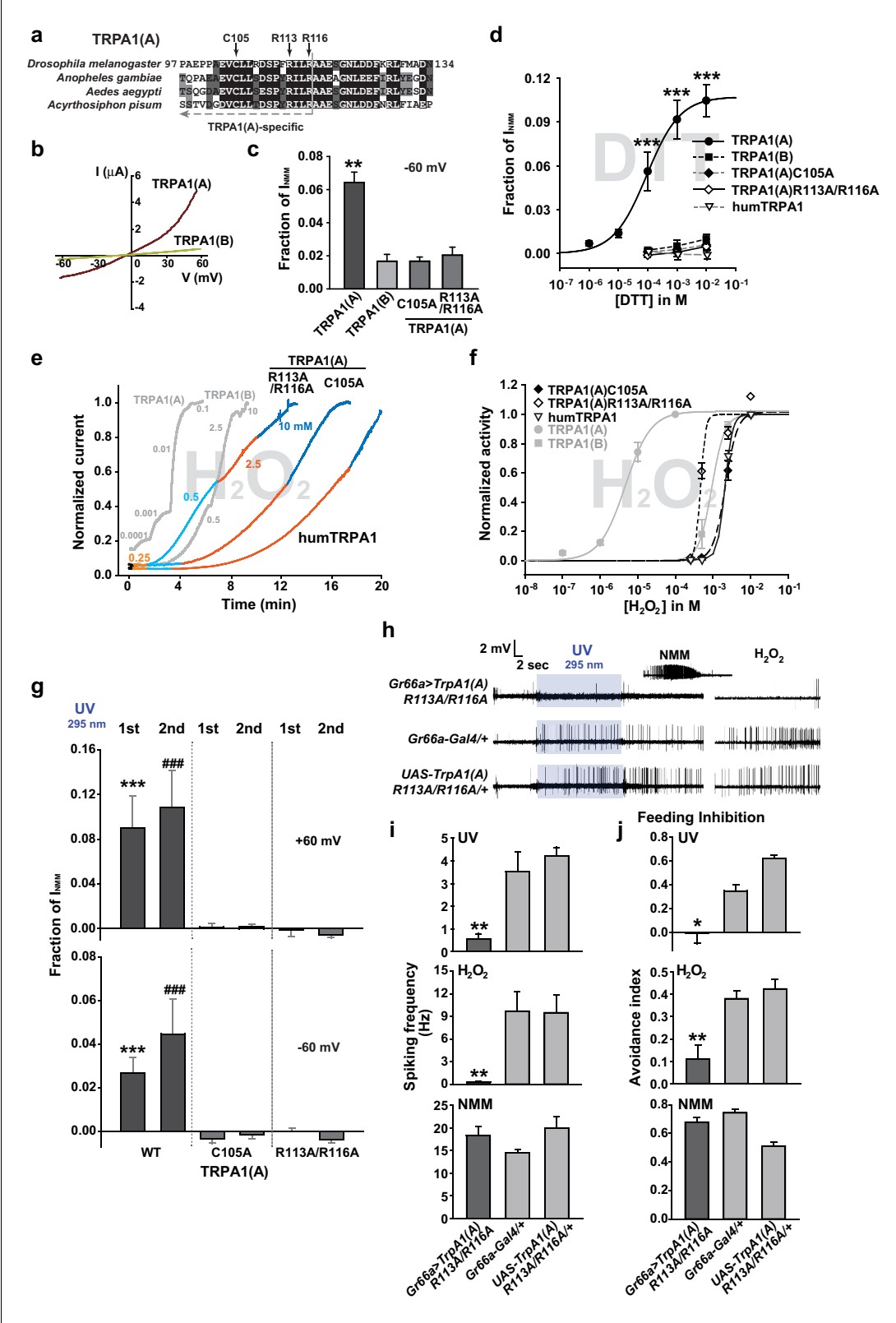

**Figure 4.** The sensitivity to nucleophilic dithiothreitol correlates with the UV and $H_2O_2$ responses of TRPA1(A). (a) Sequence alignment of TRPA1(A) N-terminal regions from indicated insects. Conserved residues that were substituted with alanine are indicated by arrows. (b) Representative current-voltage relationships in oocytes expressing TRPA1s. (c) Averaged basal activity of the indicated TRPA1 isoforms, normalized to NMM current (n = 7–37). (d) DTT dose dependence of TRPA1s, normalized to NMM current (n = 4–10). (e) Typical $H_2O_2$ responses of mutant TRPA1(A)s and human TRPA1

*Figure 4 continued on next page*

*Figure 4 continued*

(humTRPA1) in comparison with WT TRPA1(A) and TRPA1(B). (**f**) $H_2O_2$ dose dependence of TRPA1s in **e** (n = 4–6). (**g**) UV-evoked currents at +60 and −60 mV from WT and mutant TRPA1(A)s normalized to NMM current (n = 4). *** or ###: comparison among first or second responses, respectively. (**h**) Typical extracellular recordings for UV or $H_2O_2$-induced action potentials from i-bristles expressing *TrpA1(A)R113A/R116A* through *Gr66a-Gal4*. Inset: NMM response from the UV-non-responder presented underneath. (**i**) Summary of **h** (n = 6–12). Response to the electrophile NMM was unimpaired despite severe attenuation of UV and $H_2O_2$ responses upon expression of *TrpA1(A)R113A/R116A*. (**j**) Similarly to neuronal responses, feeding deterrence to UV and $H_2O_2$ was repressed by expression of *TrpA1(A)R113A/R116A* (n = 4–8). **p<0.01, *** or ###p<0.001, Tukey's test.

The following figure supplements are available for figure 4:

**Figure supplement 1.** Representative TRPA1 currents evoked by DTT in *Xenopus* oocytes.

**Figure supplement 2.** Maximal current amplitudes of TRPA1(A) by $H_2O_2$ were intermediate between those of DTT and NMM.

**Figure supplement 3.** Expression of *TrpA1(A)C105A* in bitter-tasting neurons failed to exhibit robust NMM-induced action potentials.

for UV-dependent feeding avoidance. Taken together, the nucleophile-detecting ability, which is reliant on the isoform-specific N-terminus, allows gustatory TRPA1(A) to sensitively respond to $H_2O_2$ and UV light. Thus, the TRPA1(A) N-terminus in the cytosol offers a unique activation modality that is independent of the electrophile-sensing pathway involving cysteines in the ankyrin repeat domain, which are shared between isoforms.

## Mosquito TRPA1(A) exhibits a dramatically enhanced sensitivity to nucleophilic DTT and shows elevated responsiveness to UV and $H_2O_2$ in oocytes

Next, we examined if nucleophile sensing of TRPA1(A) can be a general underlying mechanism for the detection of UV illumination in insects. TRPA1 isoforms from malaria-transmitting mosquitoes, *Anopheles gambiae* (agTRPA1), were previously reported to be similar to their *Drosophila* counterparts in thermosensitivity (*Kang et al., 2012*). The agTRPA1 isoforms were heterologously expressed in frog oocytes to investigate whether the reactions to nucleophiles, $H_2O_2$ and UV are shared characteristics of insect TRPA1(A)s. Compared to *Drosophila* TRPA1(A), cells expressing agTRPA1(A) exhibited considerably enhanced responsiveness to all three stimuli, while isoform dependencies existed, similar to *Drosophila* TRPA1 (*Figure 5* and *Supplement file 1*). DTT more robustly activated agTRPA1(A) with an order-lower EC50 of ~3.8 µM and ~10 times higher peak current amplitudes than *Drosophila* TRPA1(A) when normalized to NMM responses (*Figure 5a,b* and *Figure 5—figure supplement 1a*). Other structurally distinct nucleophiles, such as imidazole and benzyl thiocyanate (BTC), also preferentially activated agTRPA1(A) over agTRPA1(B) (*Figure 5—figure supplement 2*), demonstrating that TRPA1(A) responds to nucleophilicity and not the structures of the compounds. Oocytes microinjected with cRNA of *agTrpA1(A)* but not of *agTrpA1(B)* were often found to be of poor quality for electrophysiological characterization, presumably due to the large conductance resulting from agTRPA1(A) activation by the nucleophilic reducing power in the cytosol. To avoid this problem, 3–4 hr after cRNA microinjection, the TRPA1 antagonist ruthenium red was added to the oocyte media at a concentration of 3 µM to block the agTRPA1(A) activity induced by cytosolic reducing power and to yield cells appropriate for subsequent experiments. $H_2O_2$ elicited ~5 times larger NMM-normalized agTRPA1(A) currents with a ~3 times lower EC50 of 1.7 ± 0.3 µM than did *Drosophila* TRPA1(A) (*Figure 5c,d* and *Figure 5—figure supplement 1b*, and *Supplement file 1*). Furthermore, UV responses from agTRPA1(A) were ~5 times higher than those from *Drosophila* TRPA1(A) tested at the same settings (*Figure 5e,f*). Inside-out macropatches from oocytes showed robust UV responses when expressing agTRPA1(A) but not agTRPA1(B) (*Figure 2—figure supplement 2c–e*), which is indicative of direct UV detection. Consistent with its higher nucleophilic sensitivity and responsiveness than fly TRPA1(A), agTRPA1(A) in excised membranes exhibited little latency and larger current amplitudes than did the fly ortholog. Thus, the functional study performed using agTRPA1(A) conveys two important messages. First, detection of nucleophiles, $H_2O_2$ and UV is likely a common function of TRPA1(A) in insects. Second, nucleophile sensitivity of insect TRPA1(A) is tightly associated with the ability to rapidly detect $H_2O_2$ and UV illumination, as the sensitivities to

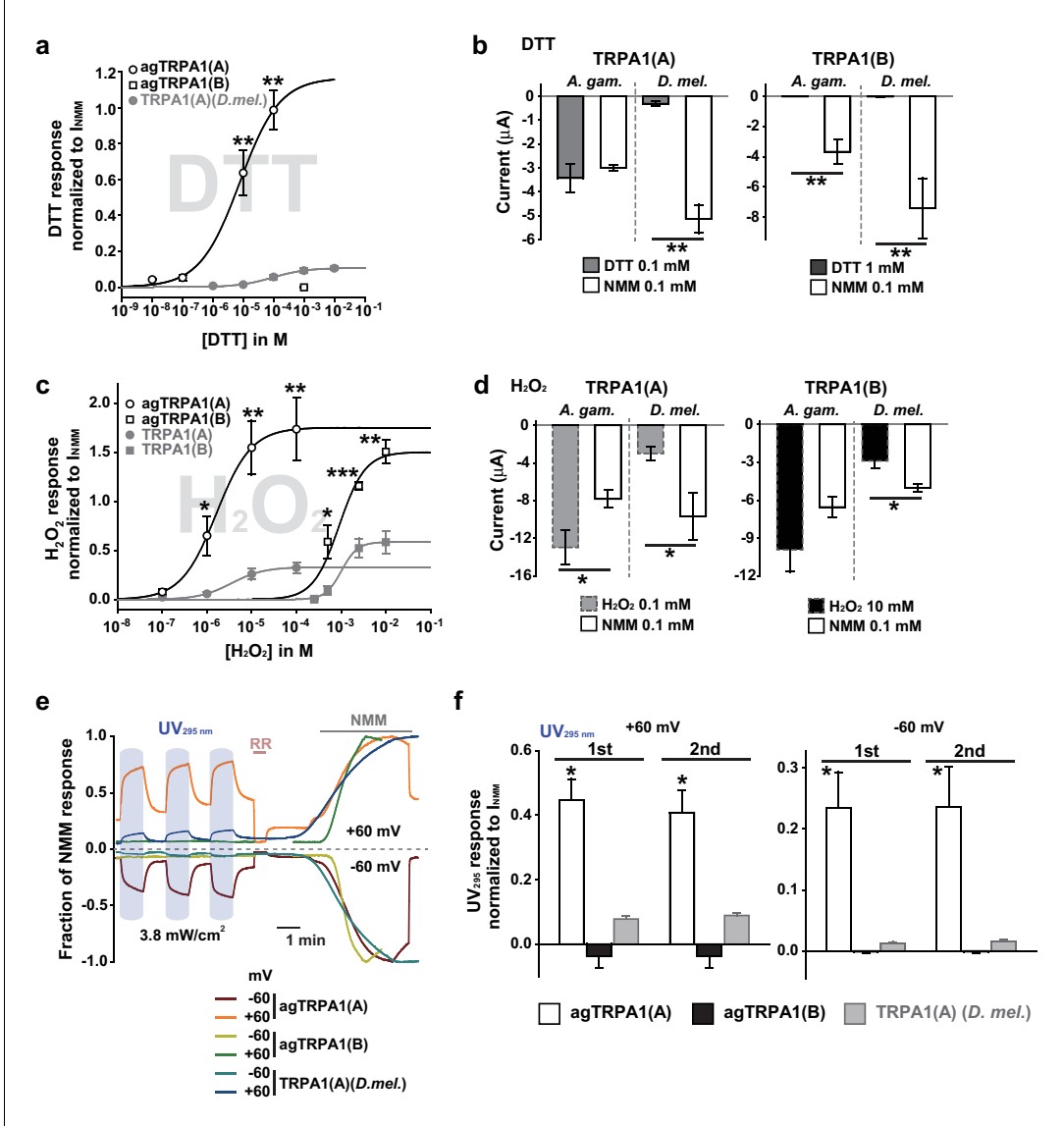

**Figure 5.** Concomitant natural variations in DTT, UV and H2O2 responsiveness between *Anopheles gambiae* and *Drosophila melanogaster* TRPA1(A)s. (a and b) In *Xenopus* oocytes, agTRPA1s show isoform dependence to DTT as do their *Drosophila* counterparts, with larger response amplitudes (n = 5–6). Dose-dependency to DTT (a) and averaged peak current amplitudes evoked by DTT and NMM (b) are presented for the channels, as indicated. (c and d) The robust DTT receptor, agTRPA1(A), exhibits enhanced $H_2O_2$ responses compared to *Drosophila* TRPA1(A) (n = 4–5). Dose-dependency to $H_2O_2$ (c) and averaged peak current amplitude (d) are compared between mosquito and fly TRPA1 isoforms. (e and f) agTRPA1(A) responds more robustly to UV light than *Drosophila* TRPA1(A), while agTRPA1(B) does not. A typical UV-evoked current response of agTRPA1(A) is superimposed on the responses of agTRPA1(B) and *Drosophila* TRPA1(A) following normalization to the NMM response (e). Normalized UV-elicited current amplitudes averaged for the indicated channels (f, n = 4–12). *$p < 0.05$, **$p < 0.01$, ***$p < 0.001$, Tukey's and Mann-Whitney U or Student's t-tests.

The following figure supplements are available for figure 5:

**Figure supplement 1.** Typical DTT (a) and $H_2O_2$ (b) responses of agTRPA1(A) and agTRPA1(B) heterologously expressed in *Xenopus* oocytes.

**Figure supplement 2.** Nucleophiles other than DTT preferentially activate TRPA1(A) over TRPA1(B).

the three stimuli are very well correlated with one another in experiments with agTRPA1(A) as well as *Drosophila* TRPA1(A)s.

## TRPA1(A) responds to natural intensities of white light in vivo and in vitro despite its suboptimal UV sensitivity

To evaluate the spectrum dependence of *TrpA1*-dependent feeding deterrence in fruit flies, monochromatic UVA light at a wavelength of 365 nm was used in the neuronal, behavioral and heterologous experiments, and the results from *Xenopus* oocytes were compared with those obtained using monochromatic UVB radiation (*Figure 6a, c, e*). WT animals showed cellular and behavioral responses to UVA which relied on *TrpA1* (*Figure 6a, c*). For robust *TrpA1*-dependent gustatory neuronal spiking, UVA at 365 nm required a much greater intensity and a longer duration of irradiation, 42.1 mW/cm$^2$ and ~1 min in total, respectively (*Figure 6a* and *Figure 6—figure supplement 1a*). *TrpA1$^{ins}$* animals were more appetitive under UVA, and consumed more sucrose than did controls, resulting in a negative avoidance index (*Figure 6c*). The behavioral deficit of *TrpA1$^{ins}$* was rescued by gustatory-specific *Gr66a-Gal4* as well as the genomic rescue transgene (*Hamada et al., 2008*; *Du et al., 2016*). Note that *wcs* show a higher avoidance than do *w$^+$* rescue flies. This is probably because the lack of eye pigments in *wcs* impairs the visual system, which is necessary for UVA attraction (*Figure 6—figure supplement 2c*; *wcs* indicated by grey boxes). The attractive nature of UVA can also be observed in the feeding deterrence assay with visually intact mini-*white*-positive *TrpA1$^{ins}$* (*Figure 6c*), as the mutants show increased ingestion upon UVA illumination. To probe the possible role of photoreceptors in feeding deterrence, the chemical synaptic transmission of photoreceptors was inhibited by the tetanus toxin light chain (TNT) expressed under the control of *GMR-Gal4*. This genetic perturbation insignificantly impaired UV-induced feeding deterrence (*Figure 6—figure supplement 2a*), while the flies failed to show typical attraction responses to UVA at 365 nm (*Figure 6—figure supplement 2b, c*). This result indicates that *TrpA1*-positive taste neurons are instrumental in avoidance, which is consistent with the suppression of feeding inhibition observed with gustatory expression of the dominant negative *TrpA1(A)* transgene (*Figure 4j*). To compare the innate sensitivity of TRPA1 isoforms to UVA and UVB light, isoforms heterologously expressed in oocytes were subjected to determination of dose dependence in response to changing light intensities (*Figure 6e*, and *Figure 6—figure supplement 1b*). Consistent with the isoform dependence of nucleophile-associated stimuli, responses to UVA were observed when TRPA1(A) but not with TRPA1(B) was expressed. The half-maximal efficacy light irradiances (EI50s) of fly TRPA1(A) to UVA and UVB were similar to each other (3.8 ± 2.2 and 2.7 ± 0.5 mW/cm$^2$ at −60 mV, respectively), although the maximal response amplitudes elicited by UVA light were relatively lower than those elicited by UVB light. UV responses of agTRPA1(A) were more robust in terms of the normalized maximal amplitude, but the EI50s (4.7 ± 2.7 and 3.0 ± 0.5 mW/cm$^2$ at −60 mV for UVA and UVB, respectively) were similar to those of fly TRPA1(A).

The total solar UV (<400 nm) intensity is ~6.1 mW/cm$^2$ (~6.8% of total solar irradiance) on the ground, and only ~0.08 mW/cm$^2$ (~1.3% of total UV irradiance) of UVB (<315 nm) reaches the ground (*RReDC*). Accordingly, the requirement of UV irradiances for the TRPA1(A)-dependent responses described above is much higher than the natural intensities of UVA or UVB light that insects receive. On the basis of this observation, it is conceivable that the *TrpA1*-dependent feeding deterrence is unlikely to occur in natural settings, although TRPA1(A) is more sensitive by far than is humTRPA1, which requires UVA intensities of ~580 mW/cm$^2$. Provided that the ability of nucleophile-detecting TRPA1(A)s to sense free radicals is the mechanistic basis of the UV responsiveness of TRPA1(A)s, we postulated that TRPA1(A) might be capable of responding to polychromatic natural sunlight, as visible light with relatively short wavelengths such as violet and blue rays is also known to generate free radicals via photochemical reactions with essential organic compounds such as flavins (*Eichler et al., 2005*; *Godley et al., 2005*). To test this possibility, *TrpA1(A)*-dependent responses were examined with white light from a Xenon arc lamp which produces a sunlight-simulating spectral output of the wavelengths higher than ~330 nm (*Figure 6—figure supplement 1c*). Less than 2% of the total spectral intensity derived from a Xenon arc lamp is UV light from 330 to 400 nm. Indeed, an intensity of 93.4 mW/cm$^2$, which is comparable to natural sunlight irradiance on the ground, substantially increased action potentials in *TrpA1*-positive taste neurons (*Figure 6b*, and *Figure 6—figure supplement 1d*). The increase in spiking was more apparent during the second 30 s illumination, while both the first and second 30 s responses to illumination required *TrpA1*. In

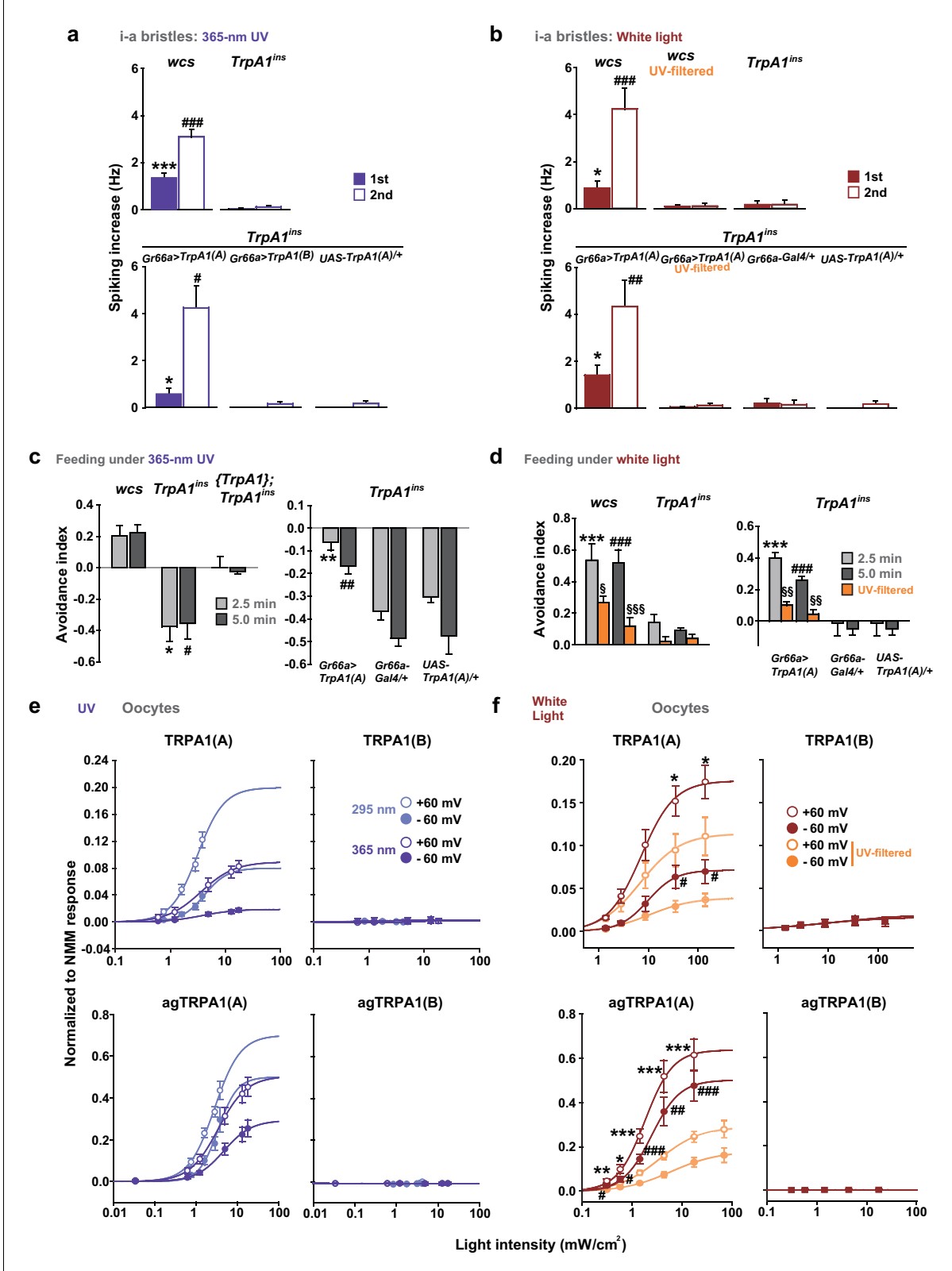

**Figure 6.** White light activates TRPA1(A) and deters feeding at natural intensities. Monochromatic UVA at 365 nm (**a**, **c** and **e**) and polychromatic white light (**b**, **d** and **f**) suppress feeding through TRPA1(A). (**a** and **b**) Illumination with 365 nm UVA light excites bitter-tasting neurons of i-a bristles at 42.1 mW/cm², and the response is dependent on *TrpA1* (n = 5–8) (**a**). Polychromatic white light from a Xenon arc lamp stimulates *TrpA1*-dependent bitter-sensing neurons at 93.4 mW/cm², which is similar to natural solar intensity (n = 5–9) (**b**). Neuronal activation by white light requires UV, as it was

*Figure 6 continued on next page*

*Figure 6 continued*

abolished upon filtering out UV with the thin titanium dioxide-coated glass. * and #p<0.05, *** and ###p<0.001, Student's t- or Tukey's test for the first and second illuminations, respectively. (c and d) UVA (c, n = 4–7) or white light illumination (d, n = 4–7) hinders feeding depending on *TrpA1(A)*. UV blocking from white light significantly reduces feeding avoidance (orange bars). * and #p<0.05, ** and ##p<0.01, *** and ###p<0.001, Tukey test for 2.5 and 5 min, respectively. §, §§ and §§§: p<0.05, 0.01 and 0.001, respectively, Student's t-tests between illuminations with and without the UV filter. (e and f) UV (e, n = 4–8) and white light (f, n = 4–8) intensity dependences of fly (upper) and mosquito (lower) TRPA1 isoforms heterologously expressed in oocytes. UVB at 295 nm (pale purple) produced higher responses than UVA at 365 nm (dark purple) (e). Blocking the UV component of white light significantly reduces the current elicited by white light illumination (f). The half maximal efficacy intensities of UV and white light are given in the text and *Supplement file 1*. Student's t-test between illuminations with and without UV filter. * and #, ** and ##, and *** and ###p<0.05, 0.01, and 0.001, respectively (*: +60 and #: −60 mV).

The following figure supplements are available for figure 6:

**Figure supplement 1.** TRPA1(A)-dependent neuronal and heterologous responses to UVA and white light.

**Figure supplement 2.** Photoreceptors are important for UVA attraction but not for UVB-dependent feeding suppression.

**Figure supplement 3.** Blue but not green light is capable of activating taste neurons, which depends on *TrpA1*.

parallel with the critical role of UV light in TRPA1(A) activation, blocking wavelengths below ~400 nm with a titanium-dioxide-coated glass filter (*Hossein Habibi et al., 2010*) (*Figure 6—figure supplement 1c*, Right) abolished the spiking responses to the level of those seen in the $TrpA1^{ins}$ neurons (*Figure 6b*). Also, polychromatic light at an intensity of 57.1 mW/cm$^2$ readily induced feeding inhibition that required *TrpA1*, and UV filtering also significantly suppressed the feeding deterrence (*Figure 6d*). In oocytes, TRPA1(A)s but not TRPA1(B)s showed current increases when subjected to a series of incrementing intensities of Xenon light (*Figure 6f*, and *Figure 6—figure supplement 1e*). Fitting the data to the Hill equation yielded EI50s of 9.8 ± 4.1 and 2.5 ± 0.7 mW/cm$^2$ for fly and mosquito TRPA1(A)s, respectively, revealing that TRPA1(A)s are sufficiently sensitive for detection of natural day light intensities. In terms of current amplitudes, agTRPA1(A) generated ~6 times more robust light-induced currents at −60 mV than did the fly ortholog isoform at the highest light intensity used. The UV filter significantly decreased the current responses, indicating the importance of UV in TRPA1(A) stimulation by white light. Furthermore, the nucleophilicity-specific mutants TRPA1 (A)C105A and TRPA1(A)R113A/R116A expressed in oocytes behaved like the nucleophile-insensitive TRPA1(B) isoform in response to white light (*Figure 6—figure supplement 1e*). These results suggest that visible light with relatively short wavelengths can substantially contribute to the excitation of *TrpA1(A)*-positive neurons, as white light from the Xenon arc lamp contains UV light at an intensity insufficient for robust activation of *TrpA1(A)*-positive taste neurons. To test this possibility, the fly labellum was illuminated with 470 nm blue light at 10 s durations at doses that were sequentially increased from 33 to 186 mW/cm$^2$, and action potentials were registered from *TrpA1*-positive i-a bristles (*Figure 6—figure supplement 3*). The serial pulses of illumination elicited spikings above the intensity of 63 mW/cm$^2$ in a *TrpA1*–dependent manner, indicating that blue light contributes to polychromatic TRPA1(A) activation in assistance of UV. In contrast, 30 sec-long illumination with green light (540 nm) rarely evoked spikings, even at a high intensity (362 mW/cm$^2$), demarcating the wavelengths capable of sufficient photochemical production of free radicals. Taken together, nucleophile sensitivity enables TRPA1(A) to detect natural solar radiation, and thus suppress feeding behavior in flies.

## UV responses of TRPA1(A) are repressed by either nucleophile or electrophile scavengers, indicating that amphiphilic free radicals are critical for light-induced TRPA1 activation

To corroborate the role of free radicals in light-induced TRPA1(A) activation, we investigated whether UV-induced TRPA1 activation could be hindered by quenching either nucleophilicity or electrophilicity, as radicals are amphiphilic. Since electrophiles react with nucleophiles, electrophilic NMM and benzyl isothiocyanate (BITC) were used as nucleophile scavengers, while the nucleophiles DTT and BTC were used as electrophile scavengers (BTC and BITC are isosteric but opposite in

chemical reactivity). Because these compounds are TRPA1(A) agonists, they are expected to increase rather than decrease TRPA1(A) activity. The agonist concentrations used were selected to be lower than those that elicit fast activation of TRPA1(A) (*Du et al., 2015*). Interestingly, pre-application of each chemical to the i-a bristles via the recording electrode lowered the frequencies of UV-evoked action potentials, regardless of scavenging polarity (*Figure 7a, b*). As *Drosophila* taste neurons may harbor multiple sensory signaling pathways, we suspected that the observed inhibition of neuronal excitation may have resulted from activation of inhibitory pathways in the bitter-tasting cells. To examine this possibility, scavenger efficacy was assessed in sweet-sensing *Gr5a-Gal4* cells exogenously expressing *TrpA1(A)*. Similar suppression of UV-induced TRPA1(A) activation was observed

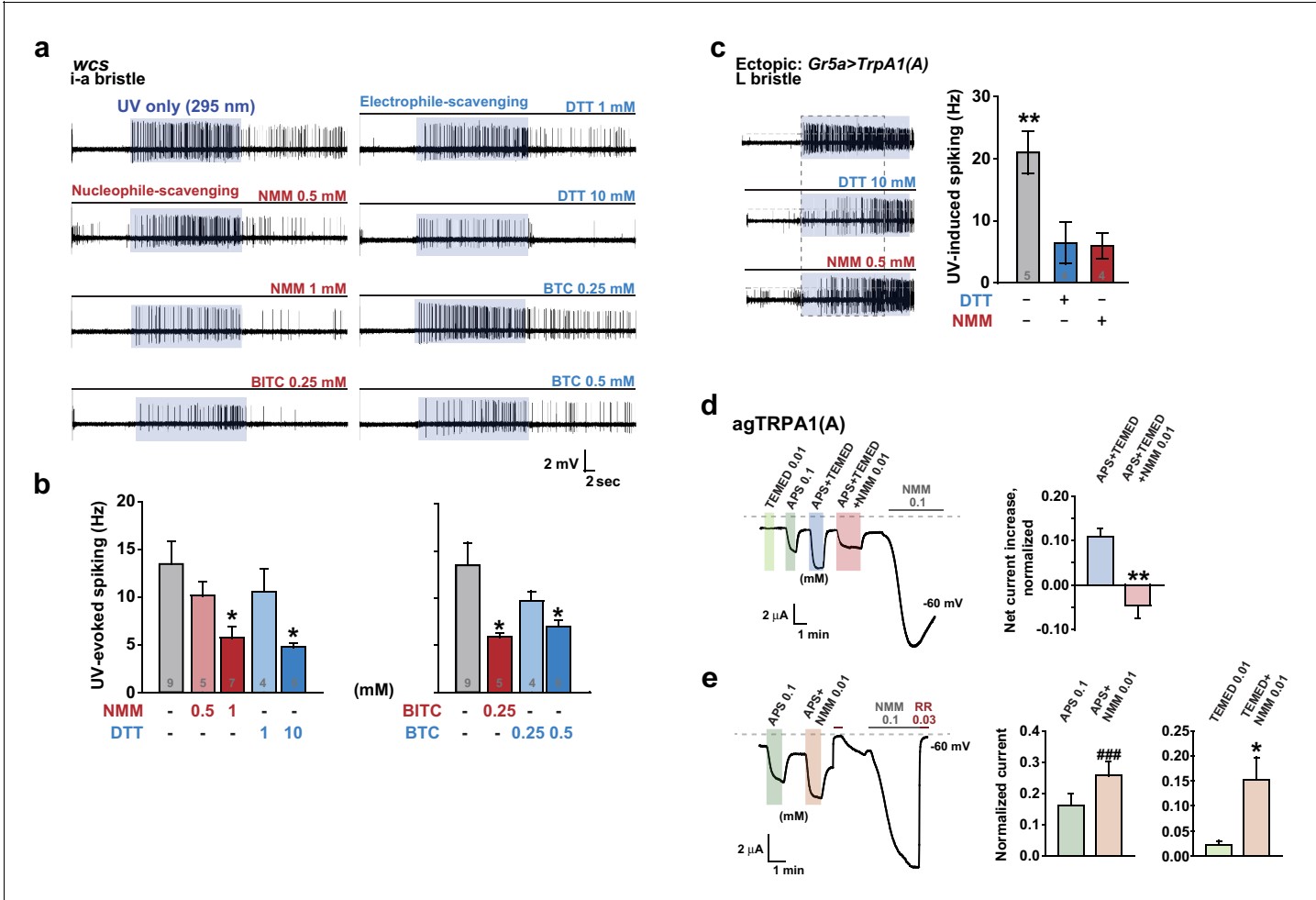

**Figure 7.** Nucleophilicity is required for UV or free radical-evoked TRPA1(A) activation. (a and b) Nucleophile-scavenging electrophiles or electrophile-scavenging nucleophiles suppress UV responses in i-a bristles. Electrophiles are in red and nucleophiles are in blue. Typical results are presented in (a), with mean and SEM values provided in (b) as bar graphs. *p<0.05, ANOVA Dunn's test. The numbers of conducted experiments are given at the bottom of each bar. (c) Ectopically expressed TRPA1(A) in sugar-sensing cells of L-bristles shows a reduced UV response in the presence of DTT or NMM. The dashed box indicates the data set that is presented in the right panel as bar graphs. **p<0.01, Tukey's test. (d) Chemically generated free radicals activate agTRPA1(A) in oocytes when ammonium persulfate (APS) and tetramethylethylenediamine (TEMED) were incubated for >30 min (see also *Figure 7—figure supplement 1*). Activation is abrogated by incubation with the nucleophile scavenger NMM. *Left*: a representative recording. *Right*: the averaged net effects of the APS/TEMED mixture on agTRPA1(A) activation with or without NMM. **p<0.01, Student's t-test (n = 4 and 6 and 6). (e) NMM does not act as an antagonist of heterologous agTRPA1(A) with either APS or TEMED alone. *Left*: a typical result with APS. *Middle* and *Right*: summary of APS (n = 5) and TEMED (n = 4–5) experiments, respectively. ###: p<0.001, paired t-test. *p<0.05, Student's t-test.

The following figure supplement is available for figure 7:

**Figure supplement 1.** TRPA1(A)-specific activation by the APS+TEMED mixture is time-dependent.

when DTT and NMM were applied in these cells (*Figure 7c*), supporting that mitigation of the TRPA1 UV responsiveness by the scavengers is unlikely to involve activation of inhibitory pathways.

However, we cannot completely rule out that, by chance, both types of taste cell share inhibitory pathways that are activated by the scavengers. Therefore, the effect of the nucleophile scavenger NMM on free radical-induced TRPA1(A) activation was tested in heterologous frog oocytes. Addition of tetramethylethylenediamine (TEMED) and ammonium persulfate (APS) initiates polymerization reactions, such as solidification of polyacrylamide gel, by generating free radicals (*Shirangi et al., 2015*). To examine the responsiveness of TRPA1(A) to free radicals, frog oocytes expressing agTRPA1(A) were exposed to a mixture of 0.01 mM TEMED and 0.1 mM APS. APS alone activated agTPRA1(A) but not agTRPA1(B) (*Figure 7d*, and *Figure 7—figure supplement 1b*), as persulfates, like peroxides, are also nucleophilic due to the alpha effect (*Edwards and Pearson, 1962*). To evaluate the net effect of radicals produced by the joint application of TEMED and APS, the cells were serially challenged in the order of 0.01 mM TEMED, 0.1 mM APS, and the TEMED and APS mixture (0.01 and 0.1 mM, respectively) (*Figure 7d*, Left). Beginning thirty minutes after mixing (*Figure 7—figure supplement 1a*), the APS/TEMED mixture activated agTRPA1(A) more robustly than did APS or TEMED alone. The 30 min latency in efficacy of the mixture is reminiscent of the incubation time necessary for solidification of a typical polyacrylamide gel after addition of APS/TEMED. Interestingly, the stimulatory effect of APS/TEMED co-incubation was abolished by adding nucleophile-scavenging NMM at 0.01 mM (*Figure 7d*). To test if NMM suppresses the action of each chemical component, either APS or TEMED was mixed with NMM for 1 hr and then applied to agTRPA1(A)-expressing cells. These experiments resulted in increases rather than decreases in the agTRPA1(A) current (*Figure 7e*), possibly reflecting the typical role of NMM as an electrophilic agonist of TRPA1 isoforms (*Kang et al., 2012*). Therefore, it is conceivable that free radicals produced by incubation of APS and TEMED activate agTRPA1(A), which is readily antagonized by nucleophile-scavenging NMM. Thus, the nucleophilic nature of amphiphilic free radicals is critical for activation of TRPA1(A), providing the mechanistic basis of light-induced feeding deterrence.

## Discussion

It is well documented that insect phytophagy is increased when UVB light is filtered out (*Bothwell et al., 1994*; *Rousseaux et al., 1998*; *Zavala et al., 2001*). The effect of UVB illumination can result from changes in plant physiology (*Kuhlmann, 2009*) or direct detection by insect herbivores (*Mazza et al., 1999*). We discovered that UV and visible light activate TRPA1(A) via a photochemical reaction that generates free radicals, thus inhibiting food ingestion by fruit flies. TRPA1(A)-expressing taste neurons appear to be responsible for feeding deterrence as light receptor cells, on the basis of three lines of evidence. First, TRPA1(A)-expressing neurons fire robustly in response to UV illumination. Second, misexpression and heterologous expression of TRPA1(A) confer light sensitivity to cells, suggesting that TRPA1(A) expression is sufficient for light responsiveness. Third, expression of a dominant negative mutant TRPA1(A) in bitter-sensing cells via *Gr66a-Gal4* eliminates light sensitivity, as assessed by feeding suppression as well as electrophysiological recordings. Because many insect genomes contain exons encoding TRPA1(A) (*Kang et al., 2012*), it would be interesting to further investigate whether TRPA1(A) expression is responsible for light sensitivity in other insects. The high responsiveness of agTRPA1(A) observed in this study implies that TRPA1(A)-dependent light detection might be a general function in insects.

Our analyses of light irradiance required for *Drosophila* feeding deterrence revealed that feeding inhibition can readily occur in response not only to UV but also to strong white light, which is likely capable of inducing nucleophilic radicals in the intracellular environment. It is conceivable that the balance between attraction by the visual system and repulsion by *TrpA1*-dependent light sensors shapes overall behavioral outcomes in natural settings under illumination with polychromatic light and that strong solar irradiation, which produces a sufficient amount of free radicals for TRPA1(A) activation, shifts the net behavioral outcomes towards repulsion. Light-induced feeding suppression is expected to occur in the middle of the day when insects are exposed to intense solar illumination. Indeed, the biting rhythm of mosquitoes is mostly out of the day time when solar irradiance is at its strongest (*Pates and Curtis, 2005*). In order to avoid harmful stimuli, animals need to overcome their urge to attractive stimuli, such as food. Feeding suppression may be a requisite for migration

to shaded places, which suggests that flies may exhibit a negative phototaxis driven by light-induced TRPA1(A) activation.

Photochemical reactions underlie rhodopsin-mediated visual mechanisms, where photon-dependent actuation of retinal covalently bound to opsin triggers a biochemical signaling cascade and an electric potential shift in the photoreceptor. We found that UV and high energy visible light, which induces photochemical generation of free radicals in the biological tissues, can be sensed without the need of a cofactor like retinal, because the basic and shared property of the radicals, such as nucleophilicity, is sensed by TRPA1(A)s. Detecting electrophilicity of reactive chemicals has been regarded as the key feature of the molecular chemical nociceptor TRPA1 in bilaterian animals (*Kang et al., 2010*), probably because of evolution of bilaterians in oxygen-rich surroundings. Because strong nucleophilicity is short-lived in the oxidative environment on Earth, animals may not have had much opportunity to adapt to the need of nucleophile detection. However, small organisms could have been under greater evolutionary pressure to develop a sensitive nucleophile-sensing mechanism. Their small size likely predisposes such organisms to be vulnerable to the effects of photochemically active light because of their high surface area-to-volume ratios, which translates into more incoming UV toxicity for a given disintoxicating capacity. The solar energy embedded in the form of light induces nucleophilicity in the cytosol while passing through the oxidizing atmosphere. We found that insects can respond to photochemically induced nucleophilicity with TRPA1(A) for sensitive and rapid detection of solar illumination. The domain for reception of nucleophilicity appears to reside in the cytoplasmic side of TRPA1(A), as the conserved residues in the cytosolic N-terminus are required for this function. Presumably, free radicals induced by photochemical reactions in the cytoplasm may remain nucleophilic longer than those in the extracellular oxidative environment due to the reducing environment within the cell. However, a receptor with a high level of nucleophile responsiveness would not well operate in this context. The cytosol is filled with reducing nucleophiles essential for redox homeostasis, which would keep the putative nucleophile receptor open and collapse the transmembrane cation gradients. Capable of synergy between the two opposing activation pathways (*Figure 8*) and tuned to conduct a limited nucleophile-dependent current, *Drosophila* TRPA1(A) is able to detect light-generated amphiphilic radicals without much disturbance from the cytosolic reducing power.

The high nucleophile responsiveness of agTRPA1(A) suggests that mosquitoes were in more need of a sensitive mechanism for nucleophile detection and, thus, probably adopted a way to suppress basal activation of TRPA1(A) by the cytosolic reducing power. In general, nucleophiles carrying extra electrons are able to form stable coordinate bonds with metal ions. Strong nucleophiles such as carbon monoxide (CO) and cyanide anions ($CN^-$) mainly exert their fatal toxicity by masking $Fe^{2+}$, which is essential for the function of heme proteins such as hemoglobins and cytochromes (*GRUT, 1954*; *Krahl and Clowes, 1940*). Thus, the differential nucleophile responsiveness between TRPA1(A)s may reflect the varying needs for avoidance due to divergent susceptibility of insects to these toxic compounds as well as strong solar irradiation. In addition, plants produce a wide variety of nucleophilic antioxidants such as phenolics, carotenoids and thiol compounds (*Pandey and Rizvi, 2009*; *Lü et al., 2010*), which suggests that nucleophile sensitivity may represent the ecological relationship of an insect species with plants. While being nectarivorous, hematophagous mosquitoes are apparently less dependent on plants for reproduction than are phytosaprophagous fruit flies (*Markow and O'Grady, 2008*). It is also plausible that mosquitoes are equipped with a heightened nucleophile detection mechanism in order to avoid dead animals when searching for a fresh blood meal, as decomposing animal carcasses emit nucleophilic gases (*Dent et al., 2004*). Therefore, the feeding niches of the species seem to be correlated with the nucleophile sensitivities of TRPA1(A)s, although it has yet to be investigated if elevated *TrpA1*-dependent nucleophile sensitivity invariably accompanies hematophagy in other insect species. Conversely, the residual nucleophile sensitivity of fly TRPA1(A) implies that the ability to detect free radical-producing light is critical to the animal, as the nucleophile responsiveness of TRPA1(A) has been evolutionarily preserved, despite the close association of *Drosophila* with plants, ever since the nucleophile sensitivity evolved in a putative common ancestor of *Drosophila* and *Anopheles*.

TRPA1(B) has been widely used as a thermogenetic tool to remotely control neurons of interest (*Bernstein et al., 2012*), and can respond to IR, which elevates the temperature of irradiated tissue (*Kang et al., 2012*). On the other hand, TRPA1(A) not only lacks thermal sensitivity for its devotion to a chemosensory role, but also detects photochemically active light such as UV light through its

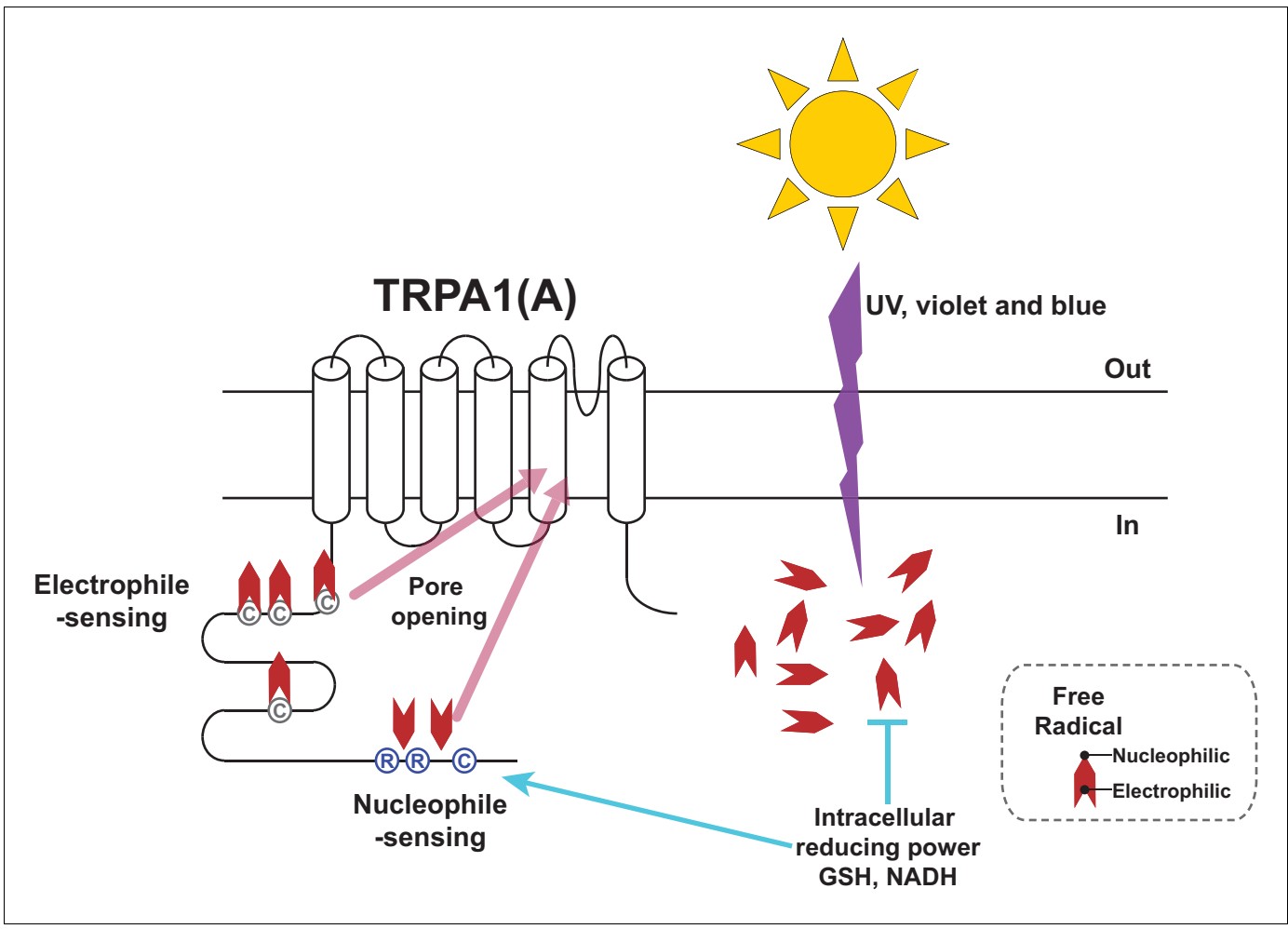

**Figure 8.** Schematic illustration of the TRPA1(A) activation mechanism in response to solar UV irradiation. Solar irradiation inflict both electrophilicity and nucleophilicity in the cytosol. The two opposing characteristics of radicals were sensed by two distinct domains of TRPA1(A). Electrophilicity is often neutralized by the cytosolic reducing power, but nucleophilicity is not interfered with. C and R in the circles respectively represent cysteine and arginine critical for each activation pathway. Chevrons depict amphiphilicity of free radicals with two opposing ends.

nucleophile sensitivity to radicals. Thus, the isoform divergence of insect TRPA1 extends its ability to perceive solar electromagnetic radiation to UV and IR, the two spectra flanking visible light. As TRPA1(A) independently serves as a UV/light sensor in nonnative cells and TRP channels show much larger single channel conductance than does channelrhodopsin-2 (*Bernstein et al., 2012*), TRPA1(A) would make an excellent optogenetic actuator that can be directly stimulated by UV light without the need for additional other proteins or chemical cofactors. Although UV radiation may be toxic to the illuminated neurons, TRPA1(A) requires a reasonable light intensity (<5.2 mW/cm$^2$: ~4% of SC and 85% of total ground UV intensity) for substantial neuronal excitation compared to the blue light intensities (2,000 to 7,500 mW/cm$^2$: 1,400 to 5,400% of SC)(*Cardin et al., 2010*) used in typical optogenetic applications. Aside from the thermal stress that likely results from intense illumination, blue light irradiation used for excitation of channelrhodopsin-2 also generates free radicals in cells at intensities as low as 2.8 mW/cm$^2$ (*Godley et al., 2005*) through ubiquitous and indispensable cellular compounds such as flavins (*Eichler et al., 2005*), and prolonged illumination with blue light can result in organismal death (*Hori et al., 2014*). Therefore, the use of TRPA1(A) in combination with low-density UV illumination might be beneficial in that TRPA1(A) may be sufficiently robust coupled with very weak but specific transcriptional promoters given its large single channel conductance. In conclusion, the nucleophile sensitivity of TRPA1(A) not only aids insects in properly responding to

solar irradiation and toxic nucleophiles, but also may potentially be used to develop a superior opto-genetic tool for neural circuitry studies in various model systems.

## Materials and methods

### Fly strains

The *UAS-TrpA1(A)* and *UAS-TrpA1(B)* transgenic lines and *TrpA1$^{ins}$* were previously described (*Kang et al., 2012*; *Rosenzweig et al., 2008*). The *UAS-TrpA1(A)C105A* and *UAS-TrpA1 (A)R113A/R116A* transgenic lines were generated by site-specific transgenesis (*Groth et al., 2004*) (Rainbow Transgenic Flies, CA, USA), inserted to the attp16 site as the *UAS-TrpA1(A)* and *UAS-TrpA1(B)* lines. The *Gr66a-Gal4* (*Dunipace et al., 2001*) and *Gr5a-Gal4* (*Marella et al., 2006*) lines were gifts from Drs. Hubert Amrein and Kristin Scott, respectively.

### Extracellular single sensillum recordings from labella bristles

In vivo taste cell recordings were performed as detailed previously (*Kang et al., 2012*). Briefly, bristles were identified based on the previously described sensillum map of the labellum (*Weiss et al., 2011*; *Tanimura et al., 2009*). The *TrpA1*-dependent response to NMM was observed in most i-bristles as reported previously (*Kang et al., 2012*). Tricholine citrate (TCC) at 30 mM was used as an electrolyte in the glass recording electrodes. Chemicals were solubilized in the electrolyte solution, and then applied to taste neurons. Spiking frequencies to chemicals were calculated for entire recordings except for $H_2O_2$ recording in L bristles, for which spiking frequencies were calculated from the first 10 s. Spike amplitudes from *Gr5a* cells expressing *TrpA1(A)* often gradually decreased to 0 mV within 20 s probably due to exhaustion of robustly firing cells. For the first 20 s of UV response recordings, the basal activity of neurons in the bristle was monitored, after which time UV illumination was administered to the sensilla for 20 s using optical fiber-coupled UV LEDs (FCS-0295–000, Mightex, CA, and UVTOP295, Qphotonics, MI, USA for UVB at 295 nm and M365FP1, Thorlabs, USA for UVA at 365 nm) controlled by an SLA-series two-channel LED driver (SLA-0100–2, Mightex) and a T-Cube LED driver (LEDD1B, Thorlab, USA), respectively. The maximal optical fiber output of 295 nm UV was 0.063 mW using a ball-lens type LED and that of 365 nm UV was 0.3 mW. These net power outputs at the tip of the optical fiber were measured with a photodiode sensor (S120VC, Thorlabs, NJ, USA) connected to a digital console (PM100D, Thorlabs, NJ, USA). Illumination intensity was calculated by considering the size of illuminated area derived from the numerical aperture (NA) values of the optical fibers and the distance to the samples. Due to the complex shape of fly taste bristles on the labellum and various illumination angles between the light beam and tissue, we simplified the calculation by postulating a 45° angle and oval illumination area at a distance (*Figure 1—figure supplement 1d*). For oocytes, circular areas were calculated (*Figure 1—figure supplement 1e*). Blue and green light illumination was accomplished using a GFP or RFP excitation filter (470 or 540 nm with a bandpass of 50, respectively) equipped with a typical fluorescence microscope. The UV filter for experiments with white light consisted of glass deposited with nanolayers of titanium dioxide (custom-made, Seoul Precision Optics, Seoul, Korea). Flies prepared for sensillum recording in response to light were used once to record from a single bristle, in order to test only naïve cells. The reference electrode containing hemolymph-like solution 3.1 (HL3.1) (*Feng et al., 2009*) was inserted close to the labella taste neuron cell bodies from the back of the fly thorax, which held the proboscis in an extended configuration in order to minimize electrical noise stemming from movement of the live animal. Tasteprobe (Syntech, Netherlands) was used as a preamplifier to register the action potentials from the neurons, which were digitized with Powerlab (ADI instruments, Australia). The obtained spiking frequencies were analyzed by Labchart (ADI instruments, Australia). Non-responding bristles were re-tested with other agonists that activate the same neurons as indicated in the main text (*Figure 1—figure supplement 2* and *Figure 3—figure supplement 1*).

### Capillary feeder assays

To quantitatively evaluate the impact of UV irradiation and chemicals on feeding deterrence, the capillary feeder (Café) assay (*Ja et al., 2007*) was used with minor modifications. In particular, feeding avoidance upon UV illumination was determined using two sibling populations of 16 hr starved

flies. One population, consisting of a vial containing 20–23 flies 2–3 days of age was illuminated with 312 nm UV light with a UV lamp (NB-UVB 311–313 nm, ATObeam, Goyang, Korea; UVB lamp, PL-S 9 W/01, Phillips, Netherlands), 365 nm UV light (LF-204.LS UVlite ultraviolet lamps, UVITEC, Cambridge, UK), or with white light from a DG4 Xenon arc lamp (Sutter, CA, USA) at a distance of 2.5 cm from the standing vial, while the other group, which had a similar number of flies, was allowed to feed freely and was left untreated at the same time (*Figure 1c*). Irradiance was measured as ~1.8, 4, and 57.1 mW/cm$^2$ for UVA, UVB, and white light, respectively, using an excised piece of a vial covering the photodiode probe (S120VC, Thorlabs, NJ, USA) to simulate internal irradiation. The vials were made of polypropylene, which has a low rate of UV transmission (*Kruenate et al., 2004*), resulting in increased internal temperature, as described in *Figure 1—figure supplement 3*. To minimize thermal accumulation, the UV-illuminated vial was actively cooled by fan-driven air flow while the internal temperature of a separately illuminated vial was concurrently monitored. After each feeding session, the change in the level of the menisci of 30 mM sucrose solutions in three calibrated glass capillary tubes (#2920107, Marienfeld, Lauda-Königshofen, Germany, 15 mm/µl) was measured. Following measurement of the evaporated volume obtained from vials without flies, the distance readings were converted to volume measurements. The ingested volume per animal was then used to calculate an 'avoidance index' by dividing [ingested volume per fly in the sucrose-only vial minus ingested volume per fly in the UV-plus-sucrose vial] by the sum of ingestion volume per fly in either vial. For the Café assay for $H_2O_2$, two capillaries containing the same solution were inserted into a vial together with two other capillaries with other tastants. The use of multiple capillaries for a single tastant mixture suppresses experimental variation, presumably owing to higher exposure of flies to tastants and an averaging effect between feeding amounts in separate tubes. To obtain an avoidance index, the volume of $H_2O_2$+sucrose consumption was subtracted from the volume of sucrose-only consumption, the result of which was in turn divided by total ingested volume.

## Proboscis extension reflex assay

The proboscis extension reflex (PER) assay was performed with modifications as previously described (*Kang et al., 2010*; *Kang et al., 2012*). UV or IR-induced PER was monitored in *TrpA1*-deficient flies expressing either *TrpA1(A)* or *TrpA1(B)* in *Gr5a-Gal4* cells. Flies that had been starved overnight were glued to glass slides, water-satiated, and illuminated with 254 nm UV light at an intensity of 0.28 mW/cm$^2$ (LF-204.LS UVlite ultraviolet lamps, UVITEC Cambridge, UK) for 2 min, during which time PER frequency was scored. When a fly fully extended its proboscis 10 times or more, a maximum score of one was given. The PER score of a fly that extended its proboscis fewer than 10 times was calculated by dividing the number of proboscis extensions by 10. For IR-evoked PER, IR from a radiant heater (940 watt, JD07010-1002, iSolar, Inchon, Korea) was administered at a distance of 20 cm from the fly.

## UV attraction behavior

UVA radiation at 365 nm was administered for 20 s from the bottom side of a horizontally placed vial (*Figure 6—figure supplement 2b*) that contained 3–4-day-old adult flies. Attraction indices were calculated by determining the fraction of the flies in the half of the vial close to the UVA source.

## Functional characterization of TRPA1 in *Xenopus* oocytes

TRPA1-dependent currents in *Xenopus laevis* oocytes induced by application of chemicals and light illumination were recorded by the two-electrode voltage clamping technique (TEVC), as described previously (*Kang et al., 2010*; *Kang et al., 2012*). Briefly, ovaries were surgically prepared and subjected to digestion with 1.5 mg/ml collagenase for 1.5 hr. Subsequently, the follicular layer of the oocytes was manually removed. One day after microinjection of 50 nl of *TrpA1* cRNA, oocytes were electrophysiologically examined while perfused with the recording solution (96 mM NaCl, 1 mM KCl, 1 mM $MgCl_2$, 5 mM HEPES, pH 7.6). For UV illumination, the optical fiber terminal was mounted above the cell at a minimal distance to achieve the highest possible intensity (*Figure 1—figure supplement 1c*). $H_2O_2$ (HP1002, GeorgiaChem, GA, USA) and DTT (43819 Sigma Aldrich, MO, USA) solutions were freshly prepared before use. For UV experiments, the initial voltage was −60 mV, and it was then changed in periods of 300 ms from −60 to +60 mV per second. For $H_2O_2$ and DTT

responses, the voltage was held constant at $-60$ mV during recording. The current was amplified with a GeneClamp 500B amplifier (Molecular Devices, CA, USA) and registered by a digitizer (Digidata 1440 A, Molecular Devices, CA, USA). Data from dose-dependence experiments were normalized with respect to 0.1 mM NMM currents recorded from the same cells, and fitted to the Hill equation using Sigmaplot12.

## Inside-out macropatch recordings

Patch-clamp recordings were carried out in an inside-out configuration using macropatches excised from *Xenopus* oocytes expressing TRPA1. Currents were recorded with an EPC 10 patch-clamp amplifier (HEKA Instruments, Germany) controlled by Patchmaster (HEKA Instruments, Germany). All current recordings were sampled at 10 kHz and filtered at 1 kHz. The patch pipettes were pulled from borosilicate capillaries (Hilgenberg-GmbH, Germany) using a Narishige puller (PC-10, Narishige, Tokyo, Japan). The patch pipettes had a resistance of 3 ~ 5 M when filled with pipette solution containing 130 mM NaOH, 3 mM HEPES, and 0.5 mM Na-EDTA adjusted to pH 7.6 with HCl. Cells were bath-perfused with a solution of 130 mM NaOH, 3 mM HEPES, and 1 mM $MgCl_2$, pH 7.6, with HCl. An oocyte was shrunk in a hypertonic solution and the vitelline membrane was removed with forceps to access the plasma membrane. All recordings were carried out at room temperature. The currents from *Xenopus* oocytes were studied by holding the potential at 0 mV and ramped from -100 to +100 mV for 500 ms and then returned to 0 mV. Currents were analyzed and fitted using Patchmaster (HEKA Instruments, Germany) and Origin6.0 (MicroCal, MA, USA).

## Statistics

To compute proper sample sizes, we used the G power program available at www.gpower.hhu.de (*Faul, 2009*). To detect differences with 80% power between the mean values of two independent groups, four replicates in each group were necessary for a Student's t-test with typical parameters (alpha = 0.05, effect size d = 3). For ANOVA Tukey's HSD tests with alpha = 0.05 and effect size f = 30, three independent samples in each group were needed to compute a difference between the mean values of two independent groups in multiple comparisons.

Student's t-tests, ANOVA Tukey's multiple comparison, ANOVA repeated measures, ANOVA Dunn's test, and Mann-Whitney U tests were performed with Sigmaplot12. Error bars indicate the standard error of mean (SEM). Normality was tested by the Shapiro-Wilk method. When data failed to pass either normality or equal variance tests, they were analyzed by rank sum tests, such as Mann-Whitney U and ANOVA Dunn's tests. Unless indicated otherwise, '\*,' '\*\*,' and '\*\*\*' or '#,' '##,' and '###' represent p values of <0.05, <0.01, and <0.001, respectively. Two groups of data were examined by Student's t- or Mann-Whitney U tests. When comparing three or more groups, ANOVA tests were used. All averaged data points and error bars represent the mean±SEM, unless indicated otherwise.

## Acknowledgements

This work was supported by Basic Science Research Program (NRF-2015R1D1A1A01057288) to KJK and Global PhD Fellowship Program (2015H-1A2A-1034723) to TJA through the National Research Foundation of Korea (NRF) funded by the Ministry of Education. We thank J H Lee for technical assistance in genetics and behavioral assays, and Drs J S Kang and S K Chung for insightful comments on the manuscript.

## Additional information

### Funding

| Funder | Grant reference number | Author |
| --- | --- | --- |
| Ministry of Education | NRF-2015R1D1A1A01057288 | KyeongJin Kang |
| Ministry of Education | 2015H-1A2A-1034723 | Tae Jung Ahn |

The funders had no role in study design, data collection and interpretation, or the decision to submit the work for publication.

## Author contributions
EJD, TJA, D-WS, HC, Conception and design, Acquisition of data, Analysis and interpretation of data, Drafting or revising the article; XW, MC, Acquisition of data, Analysis and interpretation of data, Drafting or revising the article; DLN, Conception and design, Acquisition of data, Drafting or revising the article; JYK, H-WK, Conception and design, Analysis and interpretation of data, Drafting or revising the article; KJK, Conception and design, Acquisition of data, Analysis and interpretation of data, Drafting or revising the article, Contributed unpublished essential data or reagents

## Author ORCIDs
Hana Cho, http://orcid.org/0000-0002-9394-8671
KyeongJin Kang, http://orcid.org/0000-0003-0446-469X

## Additional files

### Supplementary files
• Supplementary file 1. Response profile of TRPA1s to light and chemical agonists in *Xenopus* oocytes. Based on data acquired at −60 mV in *Xenopus laevis* oocytes, [a]Maximum activity normalized to the 0.1 mM NMM response, -: <0.01, +: 0.01 to 0.2, ++: 0.2 to 0.75, +++: >0.75, [b]N.A.: not applicable, [c]N.D.: not done, [d]Relative response normalized to the 1 mM DTT current, *: TRPA1(B) did not show current responses sufficient for fitting and EC50 estimation.

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
