## [Decision Letter]

Thank you for submitting your article "Nucleophile sensitivity of *Drosophila* TRPA1 underlies light-induced feeding deterrence" for consideration by *eLife*. Your article has been favorably evaluated by K VijayRaghavan (Senior Editor) and four reviewers, one of whom, Leslie C Griffith (Reviewer #1), is a member of our Board of Reviewing Editors and another is Simon Sprecher (Reviewer #4).

The reviewers have discussed the reviews with one another and the Reviewing Editor has drafted this decision to help you prepare a revised submission.

Summary:

This study provides very clear and detailed evidence that the (A) isoform of insect TrpA1 detects uv-generated nucleophiles at normal light levels. The channel data are nicely buttressed by behavioral studies showing uv-induced feeding suppression in insects. While the study uses uv light as its main manipulation, the results show that this isoform of TrpA1 can act as a general nucleophile detector. This is an important advance since, in contrast to electrophiles, little is known about how animals avoid the toxic effects of nucleophiles.

Essential revisions:

There are, however, a couple issues lingering from review that the authors should deal with. The main issue can be dealt with in Discussion. The technical issues are ones that require clarification rather than new experiments and addition of control data that have likely already been collected.

1) They authors should address the biological relevance of nucleophile sensitivity in both the Introduction and Discussion. Under what conditions might light-mediated feeding suppression actually occur? Do *Drosophila* (and other insects) really use TRPA1-expressing neurons as light-sensors to control feeding behaviors? While it is beyond the scope to do more experiments, providing a rationale as to why behaviors might require this aspect of dTrpA1 function would be useful to the reader.

2) The writing and flow of the paper must be improved. The manuscript is very dense and quite difficult to follow, and we strongly encourage a careful look at (and possibly thorough pruning of) the presentation and description of the results. It was frustrating to have to jump repeatedly between main and supplemental figures, which often contained different subsets of results from the same experiment. The logical links between passages were also quite unclear (for example, what motivated analysis of the mosquito TRPA1?) and the constant shifting between oocyte and in vivo experiments were sometimes confusing. A summary of the chemical properties of all the agonists/antagonists used (DTT, NMM, H2O2, UV etc.) in a brief table might also help navigation of the pharmacological analyses.

3) The authors need to provide quantitation of some of their major points. Specifically, around the issue of the similarity of rectification and reversal potential mentioned in the subsection “Heterologous expression of TRPA1(A), not TRPA1(B), in *Xenopus oocytes* provides UV-evoked current responses”. To be convincing, there should be a quantitative description of the data. In addition, the inside-out patch data would benefit from at least mention of some controls (e.g. did authors do RR to demonstrate this is not a leak current?).

---

## [Author Response]

Essential revisions:

*There are, however, a couple issues lingering from review that the authors should deal with. The main issue can be dealt with in Discussion. The technical issues are ones that require clarification rather than new experiments and addition of control data that have likely already been collected.*

*1) They authors should address the biological relevance of nucleophile sensitivity in both the Introduction and Discussion. Under what conditions might light-mediated feeding suppression actually occur? Do Drosophila (and other insects) really use TRPA1-expressing neurons as light-sensors to control feeding behaviors? While it is beyond the scope to do more experiments, providing a rationale as to why behaviors might require this aspect of dTrpA1 function would be useful to the reader.*

Background information on light-induced feeding deterrence has been inserted in the Introduction, and discussion of our findings is included while taking into account the background information in order to describe the significance of light-induced feeding suppression (Discussion). We also included a paragraph that addresses the possible roles of nucleophile sensitivity of insect TRPA1(A)s in terms of ecology and evolution in the Discussion.

Introduction: “Insects and birds are able to visualize upper-UV wavelengths (above 320 nm) via UV-specific rhodopsins (Salcedo et al. 2003; Ödeen & Håstad 2013). Visual detection of UV in this range by insects generally elicits attraction towards the UV source rather than avoidance (Craig & Bernard 1990; Washington 2010). At the same time, lower UV wavelengths, such as UVB (280-315 nm) at natural intensities, have been known to decrease insect phytophagy (Zavala et al. 2001; Rousseaux et al. 1998) via a direct effect on the animals that does not involve the visual system (Mazza et al. 1999). However, the molecular mechanism of UV-induced feeding deterrence has yet to be unraveled.”

Discussion: “It is well documented that insect phytophagy is increased when UVB light is filtered out (Bothwell et al. 1994; Rousseaux et al. 1998; Zavala et al. 2001). […] Feeding suppression may be a requisite for migration to shaded places, which suggests that flies may exhibit a negative phototaxis driven by light-induced TRPA1(A) activation.”

Discussion: “In addition, plants produce a wide variety of nucleophilic antioxidants such as phenolics, carotenoids and thiol compounds (Pandey & Rizvi 2009; Lü et al. 2010), which suggests that nucleophile sensitivity may represent the ecological relationship of an insect species with plants. […] Conversely, the residual nucleophile sensitivity of fly TRPA1(A) implies that the ability to detect free radical-producing light is critical to the animal, as the nucleophile responsiveness of TRPA1(A) has been evolutionarily preserved, despite the close association of *Drosophila* with plants, ever since the nucleophile sensitivity evolved in a putative common ancestor of *Drosophila* and *Anopheles*.”

*2) The writing and flow of the paper must be improved. The manuscript is very dense and quite difficult to follow, and we strongly encourage a careful look at (and possibly thorough pruning of) the presentation and description of the results. It was frustrating to have to jump repeatedly between main and supplemental figures, which often contained different subsets of results from the same experiment. The logical links between passages were also quite unclear (for example, what motivated analysis of the mosquito TRPA1?) and the constant shifting between oocyte and in vivo experiments were sometimes confusing. A summary of the chemical properties of all the agonists/antagonists used (DTT, NMM, H2O2, UV etc.) in a brief table might also help navigation of the pharmacological analyses.*

For better presentation, we attempted to strengthen the logical links between passages in multiple places. For each agonist we worked with, we provided a more detailed introduction on the properties of the chemical. Regarding mosquito TRPA1, it was included in our study to probe if nucleophile and UV detection by TRPA1(A) is a common function in insects. In the revised manuscript, we make this point clear as follows: “Next, we examined if nucleophile sensing of TRPA1(A) can be a general underlying mechanism for UV detection in insects”.

Interestingly, the heightened DTT sensitivity of agTRPA1(A) is well correlated with the enhanced ability of the channel to sense UV and H2O2 (correlation of gain-of-function). This observation provides strong support for the conclusion that nucleophile sensitivity underlies light detection by TRPA1(A), in addition to the loss-of-function correlation found with the *Drosophila* TRPA1(A) mutants. Besides, robust nucleophile-dependent activation of agTRPA1(A) in oocytes enabled us to present more convincing evidence that the channel is a nucleophile receptor, with APS/TEMED used for radical responsiveness as well as other agonists such as BTC and imidazole. Finally, natural variation in nucleophile sensitivity between TRPA1(A)s from fruit flies and mosquitoes offers interesting grounds for speculation into the ecological significance of nucleophile detection, which is now included in the Discussion (fourth paragraph).

We split Figure 1 into Figure 1 and 2. The introductory panels for the feeding assay in the Figure 1—figure supplement 1 have been moved to Figure 1 to eliminate the need to flip from the main to the supplementary figures. Although we considered rearranging other figures, we found that it was difficult to squeeze more figure panels in to the main figures in some cases and that other supplementary figures are better off as they are for the flow and presentation of the story.

As suggested by reviewers, a table is now included as [Supplementary-material SD1-data] that summarizes the response profile of all TRPA1s to various stimuli.

*3) The authors need to provide quantitation of some of their major points. Specifically, around the issue of the similarity of rectification and reversal potential mentioned in the subsection “Heterologous expression of TRPA1(A), not TRPA1(B), in Xenopus oocytes provides UV-evoked current responses”. To be convincing, there should be a quantitative description of the data. In addition, the inside-out patch data would benefit from at least mention of some controls (e.g. did authors do RR to demonstrate this is not a leak current?).*

We inserted the quantified data for reversal potentials registered for our oocyte recordings in the main text. Regarding quantitation of outward rectification, the ratio between conductances at two extreme voltages, 60 and -60 mV, was calculated as a rule-of-thumb analysis of rectification (subsection “Heterologous expression of TRPA1(A), not TRPA1(B), in *Xenopus oocytes* provides UV-evoked current responses”).

To test the possibility of non-specific leak current induction by UV irradiation, prolonged UV illumination (up to 350 seconds, vs. 60 seconds for the experiments shown in Figure 2—figure supplement 2) was applied to TRPA1(B)-expressing membranes, and we did not observe any current increases (please see Figure 9). To indicate this point, we state: “To exclude the possibility of leak current induced by UV illumination, we recorded from TRPA1(B)-containing membranes over extended periods of time (up to 350 seconds) and did not observe a significant increase in current.”

Author response image 1.Membrane macropatches from cells expressing TRPA1(B) isoforms fail to show current responses to prolonged UV illumination.**DOI:**
http://dx.doi.org/10.7554/eLife.18425.027